# Hybrid Policy Distillation for LLMs

**Wenhong Zhu** [1 2]  **Ruobing Xie** [3]  **Rui Wang** [1 2]  **Pengfei Liu** [1 2]

## Abstract

Knowledge distillation (KD) is a powerful paradigm for compressing large language models (LLMs), whose effectiveness depends on intertwined choices of divergence direction, optimization strategy, and data regime. We break down the design of existing KD methods and present a unified view that connects them, reformulating KD as a reweighted log-likelihood objective at the token level. We further propose `Hybrid Policy Distillation (HPD)`, which integrates the complementary advantages of forward and reverse KL to balance mode coverage and mode-seeking, and combines off-policy data with lightweight, approximate on-policy sampling. We validate HPD on long-generation math reasoning as well as short-generation dialogue and code tasks, demonstrating improved optimization stability, computational efficiency, and final performance across diverse model families and scales. The code related to this work is available at https://github.com/zwhong714/Hybrid-Policy-Distillation.

## 1. Introduction

Recent advancements in large language models (LLMs) have demonstrated remarkable performance across text generation tasks (Vaswani et al., 2017; Achiam et al., 2023), largely driven by the massive scale of parameters and high-quality data (Kaplan et al., 2020). However, such gains come at the cost of substantial computational and storage overhead (Brown et al., 2020). Consequently, compressing LLMs by reducing the number of parameters while preserving their performance has become crucial for the practical deployment of these powerful models (Gemma et al., 2025).

Knowledge distillation (KD) (Hinton et al., 2015) has emerged as a principled approach for transferring knowledge from powerful teacher models to lightweight student models. In black-box KD, the student is limited to the teacher's generation, and distillation is typically carried out via supervised fine-tuning (SFT) (Guo et al., 2025). In contrast, white-box KD can exploit access to the teacher's predictive distributions, training the student to match the teacher through distribution-level objectives such as Kullback–Leibler divergence (KLD) on logits (Kim & Rush, 2016). Recent studies on LLM distillation further emphasize the importance of selecting appropriate divergence measures for effective distribution matching (Cho & Hariharan, 2019; Mirzadeh et al., 2020; Ko et al., 2025).

However, the choice of divergence alone is insufficient (Zhang et al., 2025). The behavior of divergence-based distillation for LLMs is governed by several tightly coupled design axes, including (i) *the **direction** of divergence* (forward vs. reverse). In practice, effective distillation requires balancing the complementary inductive biases of forward and reverse KL (Binici et al., 2022; Wang et al., 2025). Forward KL (FKLD) promotes mode coverage but often yields over-smoothed predictions (Gu et al., 2023), whereas reverse KL (RKLD) emphasizes mode-seeking and distribution shaping, which can become unstable when the student–teacher gap is large (Lu & Lab, 2025). (ii) The *optimization **strategy** of the divergence* (loss vs. reward), which leads to different optimization dynamics and gradient bias (Amini et al., 2025; Shah et al., 2025). (iii) The ***data regime*** (on-policy vs. off-policy). Off-policy distillation relies on an external data source that the student learns to imitate, but it often suffers from a train–inference behavior mismatch (Agarwal et al., 2024). In contrast, on-policy distillation (OPD) samples rollouts from the student policy, which introduces a distribution shift on the teacher side for student-generated outputs and incurs substantial computational overhead (Ko et al., 2024). These factors are often intertwined, jointly shaping the distillation behavior.

Motivated by these intertwined design axes, we first formalize a reweighted log-likelihood objective to analyze SFT, FKLD, and RKLD. Under this view, FKLD and RKLD exploit the teacher's full predictive distribution to provide

---

[1]Department of Computer Science and Engineering, Shanghai Jiao Tong University [2]Shanghai Innovation Institute [3]Large Language Department, Tencent. Correspondence to: Ruobing Xie <xrbsnowing@163.com>, Rui Wang <wangrui12@sjtu.edu.cn>, Pengfei Liu <pengfei@sjtu.edu.cn>.

*Proceedings of the 43rd International Conference on Machine Learning*, Seoul, South Korea. PMLR 306, 2026. Copyright 2026 by the author(s).

dense supervision, at the cost of losing the computational efficiency of one-hot targets used in SFT. We then propose `Hybrid Policy Distillation (HPD)`, which simultaneously accounts for both divergence directions and on-/off-policy data regimes by treating divergences as token-level reward signals, while retaining the efficiency of one-hot supervision. Specifically, HPD computes token-level reweighting signals using a negative $K_1$ estimator (Schulman, 2020) applied to both offline tokens and student-sampled tokens. For offline tokens, a positive value indicates that the ground-truth token is under-estimated (i.e., the student assigns it too low a probability) and triggers forward-KL learning, whereas negative values are used to suppress over-estimation. For student-sampled tokens, over-estimation is also penalized with suppressed probability mass redistributed toward the offline expert tokens. This token-level hybridization combines the guidance of forward-and reverse-KL with the efficiency of one-hot supervision, while naturally handling both offline and lightweight on-policy sampling. We validate HPD on long-generation math reasoning as well as short-generation dialogue and code tasks, demonstrating improved optimization stability, computational efficiency, and final performance across diverse model families and scales. In summary, our contributions are as follows:

- We present a unified reweighted log-likelihood view that facilitates understanding existing distillation methods and systematically enables the development of new approaches across divergence directions, optimization strategies, and data regimes.

- We propose HPD, a token-level policy distillation method that uses reverse-$k_1$ discrepancy and prefilling-time student sampling to selectively reinforce underestimated expert actions and suppress overestimated student actions, retaining SFT-like efficiency without full-vocabulary matching or sequence rollouts.

- Experiments on math reasoning, dialogue, and code tasks show that HPD improves optimization stability and efficiency, and consistently matches or outperforms standard SFT + OPD pipelines with less cost. Moreover, HPD provides a strong initialization for OPD, leading to further performance enhancement.

## 2. Problem Formulation

**Auto-regressive Language Models.** We model next-token generation as a sequential decision process. Let $\tau = (s_1, a_1^*, s_2, a_2^*, \ldots, s_T, a_T^*)$ denotes a trajectory from an offline dataset $\mathcal{D}$, where each state $s_t$ represents the ground-truth prefix: $s_t = \mathbf{a}_{<t}^* = (a_1^*, \ldots, a_{t-1}^*)$, and $a_t^* \in \mathcal{V}$ is the expert token at step $t$ drawn from a finite vocabulary $\mathcal{V}$. A pretrained LLM induces a policy $q_\theta(a_t \mid s_t)$ parameterized

by $\theta$, which factorizes the likelihood of the expert trajectory as: $q_\theta(\tau) = \prod_{t=1}^{T} q_\theta(a_t^* \mid s_t)$. The model is pretrained via teacher forcing, by minimizing the expected negative log-likelihood (NLL) over $\mathcal{D}$ as (Ouyang et al., 2022):

$$\mathcal{L}_{\text{pretrain}}(\theta) = -\mathbb{E}_{\tau \sim \mathcal{D}} \left[ \sum_{t=1}^{T} \log q_\theta(a_t^* \mid s_t) \right]. \quad (1)$$

**Limited Expressivity of the Student Model.** The presence of multiple valid continuations, together with label smoothing, results in an inherently soft supervision signal (Zhu et al., 2024). Consequently, in highly complex output spaces, the teacher distribution or the offline data distribution often exhibits pronounced multimodality, exceeding the modes a capacity-limited student can represent (Gu et al., 2023). Accordingly, distillation dilutes probability mass across too many modes, ultimately harming generation quality.

## 3. Preliminary

We first review SFT as a special case of KD. While SFT provides sparse supervision on target tokens, KD leverages the teacher's full predictive distribution to offer denser learning signals. However, exact divergence minimization is intractable in large-vocabulary settings, necessitating practical approximations such as Monte Carlo (MC) estimation.

### 3.1. KD via SFT

SFT trains a model on an offline dataset $\mathcal{D} = \{(s_t, a_t^*)\}$ by minimizing the cross-entropy loss. Specifically, the supervision signal is represented as a one-hot distribution $\mathbf{y}_t \in \{0, 1\}^{|\mathcal{V}|}$, where $y_{t,a_t^*} = 1$ and $y_{t,a} = 0$ for all $a \neq a_t^*$. The resulting objective is

$$\min_\theta \mathcal{L}^{\text{SFT}}(\theta) = \min_\theta -\mathbb{E}_s \mathbb{E}_{a \sim \mathcal{D}} \left[ \log q_\theta(a|s) \right]. \quad (2)$$

While effective and stable, SFT provides supervision only at a single target action, yielding a sparse learning signal that ignores alternative plausible actions.

### 3.2. KD via KL Divergence

By introducing the teacher distribution level signal, we can minimize a predictive divergence $\mathbb{D}$ between the teacher distribution and the parameterized student distribution. The common use is FKLD and RKLD.

**FKLD.** Forward KL measures how well the student distribution $q_\theta(a \mid s)$ covers the modes of the teacher distribution $p(a \mid s)$, defined as follows (Kim & Rush, 2016):

$$\min_\theta \mathbb{D}_{\text{KL}}(p\|q_\theta) = \min_\theta \mathbb{E}_s \mathbb{E}_{a \sim p(\cdot|s)} \left[ \log \frac{p(a|s)}{q_\theta(a|s)} \right]. \quad (3)$$

Recent studies (Agarwal et al., 2024; Ko et al., 2024), approximate the distribution matching by minimizing Equa-

tion (4), assuming that the teacher's distribution is similar to its training dataset $\mathcal{D}$ for the sake of efficiency, which can be decomposed into a sum of token-wise distillation:

$$
\min_\theta \frac{1}{|\mathcal{D}|} \sum_{(\boldsymbol{s},\boldsymbol{a}) \in \mathcal{D}} \left[ p(\boldsymbol{a}|\boldsymbol{s}) \log \frac{p(\boldsymbol{a}|\boldsymbol{s})}{q_\theta(\boldsymbol{a}|\boldsymbol{s})} \right]
$$

$$
= \min_\theta \frac{1}{|\mathcal{D}|} \sum_{(\boldsymbol{s},\boldsymbol{a}) \in \mathcal{D}} \sum_t^{|\boldsymbol{a}|} \left[ \sum_{a_t \in \mathcal{V}} p(a_t|s_t) \log \frac{p(a_t|s_t)}{q_\theta(a_t|s_t)} \right] \tag{4}
$$

The gradient of Equation (3) is as follows:

$$
\nabla_\theta \mathbb{D}_{\mathrm{KL}}(p\|q_\theta) = -\mathbb{E}_{\boldsymbol{s}} \mathbb{E}_{\boldsymbol{a} \sim p(\cdot|\boldsymbol{s})} \left[ \nabla_\theta \log q_\theta(\boldsymbol{a} \mid \boldsymbol{s}) \right] \tag{5}
$$

It encourages the student to acquire new knowledge by covering all behaviors that the teacher considers likely, as it penalizes missing modes heavily (Song et al., 2020). However, if the student model lacks sufficient capacity to match the full support of the teacher distribution, it may exhibit an overly smooth distribution in an attempt to cover all possible modes (Gu et al., 2023; Wang et al., 2023).

**RKLD.** Reverse KL encourages the student to focus on the high-probability modes of the teacher, potentially ignoring less likely outputs, calculated as follows:

$$
\min_\theta \mathbb{D}_{\mathrm{KL}}(q_\theta\|p) = \min_\theta \mathbb{E}_{\boldsymbol{s}} \mathbb{E}_{\boldsymbol{a} \sim q_\theta(\cdot|\boldsymbol{s})} \left[ \log \frac{q_\theta(\boldsymbol{a}|\boldsymbol{s})}{p(\boldsymbol{a}|\boldsymbol{s})} \right]. \tag{6}
$$

The gradient is as follows:

$$
\nabla_\theta \mathbb{D}_{\mathrm{KL}}(q_\theta\|p) = \mathbb{E}_{\boldsymbol{s}} \mathbb{E}_{\boldsymbol{a} \sim q_\theta(\cdot|\boldsymbol{s})}
$$
$$
\left[ \nabla_\theta \log q_\theta(\boldsymbol{a} \mid \boldsymbol{s}) \cdot (\log q_\theta(\boldsymbol{a} \mid \boldsymbol{s}) - \log p(\boldsymbol{a} \mid \boldsymbol{s})) \right]. \tag{7}
$$

This gradient structure biases the student toward the teacher's dominant modes, underrepresenting low-probability but valid outputs (Wang et al., 2025), and can further lead to unstable training when the student is poorly aligned with the teacher due to high-variance gradients induced by the unbounded log-ratio (Ko et al., 2024).

### 3.3. Monte Carlo KLD Approximation

Exactly computing the KLD is often intractable, as shown in Equation (4), since it requires summing over the full vocabulary, sequence length, and batch size. Instead, the KLD can be approximated via MC sampling. The simplest such estimator, denoted as $K_1$, at a given time step $t$, is

$$
K_1 \triangleq \frac{1}{N} \sum_{i=1}^N \log \frac{q_\theta(a_t^{(i)} \mid s_t)}{p(a_t^{(i)} \mid s_t)}, \quad a_t^{(i)} \sim q_\theta(\cdot \mid s_t), \tag{8}
$$

which is an unbiased estimator of $\mathbb{D}_{\mathrm{KL}}(q_\theta\|p)$ but suffers from high variance, as the log-ratio term is negative for a substantial fraction of samples (Schulman, 2020).

*Table 1.* Comparison of classical SFT and KD methods.

| Method | Data source | Reweighted term $w(a_t \mid s_t)$ |
|---|---|---|
| SFT | Off-policy ($\mathcal{D}$ or $\mathcal{D}^{\pi_T}$) | $\mathbf{1}[a_t = a_t^*]$ |
| FKLD | Off-policy ($\mathcal{D}$ or $\mathcal{D}^{\pi_T}$) | $p(a_t \mid s_t)$ |
| RKLD | On-policy ($\mathcal{D}^{\pi_\theta}$) | $\log p(a_t \mid s_t) - \log q_\theta(a_t \mid s_t)$ |

$K_1$ **Estimator as Reward.** Variant estimators of the KLD can be integrated into training either as a token-level reward penalty, as in PPO (Schulman et al., 2017), or as an explicit loss term, as popularized by GRPO (Shao et al., 2024). Recently, the OPD framework computes negative $K_1$ by evaluating the teacher's log-probabilities on tokens $a_t$ sampled from the student policy (Lu & Lab, 2025) as a reward. This approach enables efficient distillation, provides an unbiased gradient estimator, and improves training stability (Shah et al., 2025). See Appendix B.1 for derivation.

## 4. Our Method: HPD

### 4.1. Towards a Unified View of Distillation

From the perspective of reweighted likelihood optimization, SFT and KL distillation objectives can be unified as:

$$
\mathcal{L}(\theta) = \min_\theta -\mathbb{E}_{(s_t, a_t) \sim \mathcal{D}^\pi} \left[ w(a_t \mid s_t) \log q_\theta(a_t \mid s_t) \right], \tag{9}
$$

where $\mathcal{D}^\pi$ denotes the data source: for on-policy methods, $\mathcal{D}^{\pi_\theta}$ is sampled from the current student policy; for off-policy methods, $\mathcal{D}^\pi$ comes from a fixed dataset $\mathcal{D}$ or from a teacher policy $\mathcal{D}^{\pi_T}$. The reweighted term $w(a_t \mid s_t)$ captures a local discrepancy between the student and teacher policies at step $t$, quantifying how the probability assigned to action $a_t$ by the student deviates from that of the teacher at state $s_t$. For clarity, the specific choices of $w(a_t \mid s_t)$ for different objectives are summarized in Table 1.

**Effectiveness of Positive and Negative Optimization.** By inspecting Equation (9), we observe that positive weights increase the likelihood of the corresponding actions, whereas negative weights suppress them. The resulting gradient on a sampled token $a_t$ can be expressed as:

$$
-\frac{\partial \mathcal{L}(\theta)}{\partial z_v} \propto \begin{cases} \hat{w}_t \cdot q_v \cdot (1 - q_v), & \text{if } v = a_t, \\ -\hat{w}_t \cdot q_{a_t} \cdot q_v, & \text{if } v \neq a_t, \end{cases} \tag{10}
$$

where $z_v$ denotes the logit associated with token $v$ and $\hat{w}_t$ abbreviates $w(a_t \mid s_t)$. See Appendix D for derivation. This formulation shows that, although the objective directly targets a single token, the induced gradients propagate across the entire predictive distribution. In particular, when $\hat{w}_t < 0$, the gradient actively suppresses the sampled token and redistributes probability mass to alternative tokens proportional to their current probabilities. As discussed in Section 3.3, the reverse $K_1$ estimator naturally exhibits this behavior.

## 4.2. Hybrid Policy Distillation

We define *expert token* as the token from the teacher generation (Kim & Rush, 2016) or offline ground truth that approximates the teacher distribution (Hinton et al., 2015), while a *non-expert token* is sampled from the student.

**Hybrid Forward–Reverse KL.** Given a fixed offline expert dataset $\mathcal{D} = \{(s_t, a_t^*)\}$, we adopt the **negative** reverse $k_1$ estimator:

$$k_1 = q_\theta(a_t^*|s_t)\big[\log p(a_t^*|s_t) - \log q_\theta(a_t^*|s_t)\big], \quad (11)$$

to measure the gap between the teacher and the student on the expert token $a_t^*$ conditioned on state $s_t$. A positive $k_1 > 0$ indicates that the student underestimates the likelihood of the expert token. To correct this, we incorporate the reweighted term $p(a_t^*|s_t)$ and define the expert token weight $w_t^*$ as:

$$w_t^* \leftarrow \begin{cases} p(a_t^*|s_t) + k_1, & \text{if } k_1 > 0, \\ k_1, & \text{if } k_1 \le 0. \end{cases} \quad (12)$$

This design can be interpreted as a hybrid of forward- and reverse-KL distillation implemented via a reweighting mechanism. Unlike the weighted-sum divergence approach, which combines the two KL directions with fixed coefficients, we introduce a **masking** mechanism: when $k_1 \le 0$, the corresponding forward-KL weight is masked, reflecting that the student already overestimates the expert token and thus preventing conflicting gradient directions.

**Hybrid Policy.** To improve distribution alignment under offline autoregressive prefixes, we let the student **sample** a different token: $a_t \sim q_\theta(\cdot|s_t)$, s.t. $a_t \ne a_t^*$, given the ground-truth offline prefix $s_t$, and compute its $k_1$ gap estimator as shown in Equation (11) on that sampled token named $k_1'$. We then mask positive values of $k_1'$ by setting

$$w_t \leftarrow \begin{cases} 0, & \text{if } k_1' \ge 0, \\ k_1', & \text{if } k_1' < 0. \end{cases} \quad (13)$$

which prevents reinforcing non-expert tokens while suppressing unreasonable sampling.

As analyzed in Section 4.1, a negative weight redistributes probability mass across the vocabulary in proportion to the current model distribution. When $k_1' < 0$, to explicitly encourage the expert action, we apply a **reinforcement** to the expert token by assigning it a doubled forward-KL weight when $k_1 > 0$, and maintain a forward-KL weight when $k_1 = 0$, yielding the following expert weight:

$$w_t^* \leftarrow \begin{cases} 2p(a_t^*|s_t) + k_1, & \text{if } k_1 > 0 \text{ and } k_1' < 0 \\ k_1, & \text{if } k_1 < 0. \\ p(a_t^*|s_t) + k_1, & \text{otherwise.} \end{cases} \quad (14)$$

Finally, the two weights are integrated into reweighted likelihood optimization by binding them to their respective tokens:

$$\mathcal{L}_{\text{HPD}} = \min_\theta \mathbb{E}_{(s_t, a_t^*) \sim \mathcal{D}, a_t \sim q_\theta(\cdot|s_t)} \\ [-w_t^* \log q_\theta(a_t^* \mid s_t) - w_t \log q_\theta(a_t \mid s_t)]. \quad (15)$$

**Intuitive Explanation of HPD.** We illustrate the training procedure in Algorithm 1. HPD adopts an asymmetric token-level design. For each offline state $s_t$, the expert token $a_t^*$ is the primary optimization, explicitly aligned with the teacher, while suppressing unnecessary learning that could lead to overestimation of the expert token. A student-sampled non-expert token $a_t \sim q_\theta(\cdot \mid s_t)$ is introduced only to identify and suppress unreasonable model behavior. When the student underestimates the expert token ($k_1 > 0$), the expert token is reinforced via a forward-KL weight. This weight is further strengthened when the student overestimates a sampled non-expert token ($k_1' < 0$), ensuring that suppressed probability mass is redirected toward the expert action.

---

**Algorithm 1** HPD Algorithm

---

1: **input** student $q_\theta$, teacher $p$, dataset $\mathcal{D}$
2: **output** updated parameters $\theta$
3: Sample offline trajectories $\mathcal{T} \sim \mathcal{D}$
4: **for** each $(s_t, a_t^*) \in \mathcal{T}$ **do**
5:     Compute log-probabilities:
6:       $(\log q^*, \log p^*) \leftarrow (\log q_\theta, \log p)(a_t^*|s_t)$
7:     Compute expert reverse-KL gap:
8:       $k_1 \leftarrow q_\theta(a_t^*|s_t)(\log p^* - \log q^*)$
9:     Sample $a_t \sim q_\theta(\cdot|s_t)$
10:    Compute sampled-token reverse-KL gap:
11:      $k_1' \leftarrow q_\theta(a_t|s_t)\big[\log p(a_t|s_t) - \log q_\theta(a_t|s_t)\big]$
12:    Compute expert weight:
13:      $w_t^* \leftarrow \begin{cases} 2p(a_t^*|s_t) + k_1, & k_1 > 0, \ k_1' < 0 \\ k_1, & k_1 < 0 \\ p(a_t^*|s_t) + k_1, & \text{otherwise} \end{cases}$
14:    Compute sampled-token weight:
15:      $w_t \leftarrow \mathbb{I}[a_t \ne a_t^*] \cdot \mathbb{I}[k_1' < 0] \cdot k_1'$
16: **end for**
17: Update parameters:
18:    $\theta \leftarrow \theta - \alpha \nabla_\theta \mathcal{L}_{\text{HPD}}$

---

## 5. Experiments

In this section, we investigate distillation on long- and short-text generations. We evaluate long-generation distillation on mathematical reasoning tasks and short-generation distillation on dialogue and code tasks. Additional experimental results are presented in Section 8. To ensure a fair comparison, we separately consider offline and on-policy data.

**Baselines.** From a unified perspective, all baselines consid-

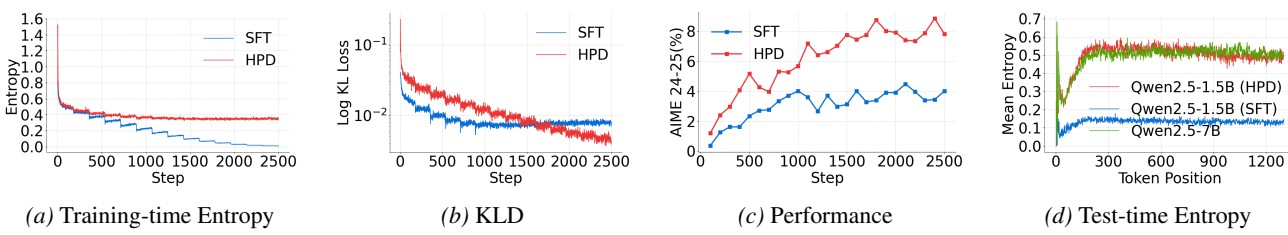

| (a) Training-time Entropy | (b) KLD | (c) Performance | (d) Test-time Entropy |

*Figure 1.* Comparison of training dynamics between SFT and HPD.

*Table 2.* Detailed results of the **off-policy data for reasoning**. For AIME and AMC, the results are avg.@32. For the other benchmarks, the results are avg.@8. ∗ denotes that the performance is statistically significant (t-test with $p < 0.01$).

| | Qwen 2.5 (7B → 1.5B and 3B) | | | | | | | LLaMA 3 (8B → 1B and 3B) | | | | | | |
|---|---|---|---|---|---|---|---|---|---|---|---|---|---|---|
| Method | AIME24 | AIME25 | AMC | Math | Obly. | GPQA | Avg. | AIME24 | AIME25 | AMC | Math | Obly. | GPQA | Avg. |
| $\mathcal{M}_T$ | 28.13 | 27.19 | 71.72 | 87.48 | 58.50 | 43.43 | 52.74 | 14.27 | 18.02 | 55.23 | 77.78 | 47.74 | 36.23 | 41.55 |
| $\mathcal{M}_S$ | 2.19 | 1.04 | 21.17 | 46.78 | 16.52 | 23.04 | 18.46 | 0.73 | 0.10 | 8.98 | 24.93 | 5.02 | 9.41 | 8.20 |
| SFT | 2.81 | 6.04 | 28.83 | 55.25 | 24.87 | 19.02 | 22.80 | 0.83 | 1.04 | 17.34 | 33.30 | 12.52 | 18.24 | 13.88 |
| SeqKD | 5.31 | 5.31 | 33.83 | 60.28 | 29.48 | 23.42 | 26.27 | 0.42 | 0.94 | 21.09 | 36.45 | 12.94 | 20.01 | 15.31 |
| RKLD | 5.00 | 3.85 | 34.45 | 58.78 | 27.41 | 27.40 | 26.15 | 0.42 | 0.63 | 17.81 | 34.10 | 12.56 | 23.86 | 15.07 |
| JSD | 5.73 | 4.90 | 35.31 | 59.63 | 27.30 | 25.69 | 26.43 | 0.31 | 0.83 | 18.98 | 35.95 | 14.72 | 22.92 | 15.62 |
| **HPD** | **7.71**∗ | **9.89**∗ | **39.84**∗ | **63.40**∗ | **32.53**∗ | **28.09**∗ | **30.24** | **1.25**∗ | **2.08**∗ | **21.80**∗ | **41.73**∗ | **17.24**∗ | **23.93** | **18.01** |
| $\mathcal{M}_S$ | 6.67 | 2.50 | 38.20 | 64.08 | 28.17 | 29.86 | 28.25 | 7.08 | 0.73 | 23.35 | 46.95 | 15.11 | 23.35 | 19.43 |
| SFT | 10.10 | 12.60 | 46.33 | 69.78 | 36.89 | 20.33 | 32.67 | 8.13 | 10.73 | 41.25 | 65.23 | 33.69 | 17.67 | 29.45 |
| SeqKD | 11.56 | 14.48 | 47.66 | 74.48 | 40.48 | 24.62 | 35.55 | 9.06 | 12.71 | 42.89 | 67.25 | 36.78 | 22.35 | 31.84 |
| RKLD | 9.38 | 12.29 | 46.25 | 69.58 | 37.35 | 19.51 | 32.39 | 7.19 | 6.56 | 42.66 | 65.63 | 33.48 | 26.96 | 30.41 |
| JSD | 10.31 | 14.90 | 50.70 | 73.88 | 40.69 | 29.17 | 36.61 | 6.67 | 7.40 | 41.56 | 64.75 | 33.65 | 25.69 | 29.95 |
| **HPD** | **13.75**∗ | **18.13**∗ | **54.14**∗ | **76.30**∗ | **45.33**∗ | **31.31**∗ | **39.83** | **10.94**∗ | **12.71** | **48.28**∗ | **69.25**∗ | **39.02**∗ | **27.15**∗ | **34.56** |

ered in this work can be interpreted as approximations of a reweighted log-likelihood objective with different choices of weighting estimators. See Appendix E for explanation. In practice, directly optimizing the full KL objective under long auto-regressive generation is often prohibitively expensive in terms of both memory and computation (Zhang et al., 2025). For the off-policy data, we approximate several representative methods as follows: SFT with a constant estimator 1; SeqKD (Kim & Rush, 2016) with the teacher probability $p$ as the estimator; RKLD (Gu et al., 2023) using the estimator $q \cdot (\log q - \log p)$; and JSD (Agarwal et al., 2024) using the estimator $\frac{1}{2}q \cdot (\log q - \log \frac{p+q}{2})$. See Appendix C for derivation. For the on-policy data, we follow prior work and study the effect of different initializations under the OPD framework (Gu et al., 2023; Lu & Lab, 2025; Agarwal et al., 2024).

### 5.1. Off-policy Data for Reasoning

#### 5.1.1. SETUP

*(1) Models and Datasets.* Our student models are drawn from two model families with varying parameter scales: the Qwen2.5 series (1.5B, 3B, and 7B) (Yang et al., 2025) and the LLaMA3 series (1B, 3B, and 8B) (Dubey et al., 2024). For each model family, we select the largest model as the teacher. We focus on the mathematics domain to enhance

general LLM capabilities via long chain-of-thought (CoT) reasoning. Specifically, we employ the OpenR1-Math-8192 dataset (Face, 2025). *(2) Implementation Details.* For generation efficiency, recent works (e.g., Ko et al. (2024); Agarwal et al. (2024)) often approximate the distribution under the assumption that the teacher distribution closely aligns with the empirical distribution of its training dataset. Accordingly, we first train the teacher model on the offline dataset and subsequently refine it using GRPO. Student models are fine-tuned for approximately 2k steps with a batch size of 256, and the final checkpoint is selected based on validation performance. Additional implementation details are provided in Appendix F.

#### 5.1.2. RESULTS AND ANALYSIS

**HPD achieves continual distillation.** As shown in Figure 1a, during training, SFT causes student models to quickly overfit the dataset, resulting in an entropy collapse, whereas HPD maintains stable entropy without such collapse. Similarly, as illustrated in Figure 1b, HPD progressively reduces the gap between the student and teacher distributions, while the generation performance steadily improves throughout training ( Figure 1c). In contrast, under SFT, the KL divergence gap stagnates, and the model's performance shows little to no improvement. HPD enables the

student model to gradually align with the teacher's distribution, continuously enhancing its performance toward that of the teacher, up to the inherent capacity limits of the student model.

**HPD achieves train–inference behavior alignment.** We select 1,000 prompts from the validation set and track the inference-time entropy dynamics, averaging the entropy at each token position, as shown in Figure 1d. The results indicate that, with HPD, the student model closely aligns with the teacher model across both training and inference, demonstrating consistent behavior between the two stages.

**HPD achieves great distillation performance.** As we can see from Table 2, distillation methods generally outperform SFT, showcasing their potential. The results show that HPD consistently outperforms all baseline distillation methods across both Qwen 2.5 and LLaMA 3 model families. Most notably, HPD empowers the 3B variants to reach reasoning capabilities comparable to larger models—it improves Qwen 2.5 3B by **41.0%** (from 28.25 to 39.83) and LLaMA 3 3B by **77.9%** (from 19.43 to 34.56).

## 5.2. Off-policy Data for Personalization

**Setup.** *(1) Model and Dataset.* Unlike the long-CoT reasoning task in Section 5.1, which requires extensive offline generation computation, in this section, we use the Instruct version of the corresponding model and generate distillation data by prompting it with examples from the Ultrafeedback dataset (Cui et al., 2023). *(2) Implementation Details.* All evaluation checkpoints are selected from student models fine-tuned for 2k steps with a batch size of 256. *(3) Evaluation.* See Appendix G.

*Table 3.* Detailed results of the **off-policy data for personalization** for Qwen2.5 (7B → 1.5B). We report performance on AlpacaEval2 with length control (AE-LC) and winning rate (WR), as well as on Arena-Hard and MT-Bench multi-turn dialogue benchmarks (MT-1T and MT-2T).

| Method | AE-LC(%) | AE-WR(%) | Arena-WR(%) | MT-1T | MT-2T |
|---|---|---|---|---|---|
| $\mathcal{M}_T$ | 36.04 | 34.95 | 60.00 | 9.00 | 7.44 |
| $\mathcal{M}_S$ | 8.67 | 7.47 | 9.90 | 6.64 | 5.00 |
| SFT | 12.74 | 13.72 | 18.10 | 6.80 | 4.81 |
| SeqKD | 7.83 | 9.51 | 15.40 | 6.24 | 4.15 |
| RKLD | 11.26 | 12.00 | 17.80 | 6.96 | 5.19 |
| JSD | 13.48 | 13.89 | 20.20 | 6.96 | 5.21 |
| **HPD** | **13.75** | **14.25** | **21.80** | **7.23** | **5.84** |

**Results and Analysis.** As shown in Table 3, among all distillation methods, HPD stands out as the most effective, with a particularly strong advantage in preserving multi-turn dialogue capabilities. It achieves the generally highest scores on MT-1T and MT-2T tasks, demonstrating its superior ability to retain conversational coherence and contextual understanding. HPD also leads in key alignment

performance metrics like AE-LC, AE-WR, and Arena-WR, further validating its robustness.

## 5.3. Off-policy Data for Coding

**Setup.** *(1) Model and Dataset.* We use Qwen2.5-Coder-7B-Instruct (Hui et al., 2024) and DeepSeek-Coder-6.7B-Instruct (Guo et al., 2024) as teacher models, and Qwen2.5-Coder-1.5B and DeepSeek-Coder-1.3B as the corresponding student models. For distillation, we employ the Instruct version of each model and generate training data by prompting with examples from the WizardCoder dataset (Luo et al., 2024). *(2) Evaluation.* Evaluation is performed using the EvalPlus framework (Liu et al., 2023), with greedy decoding on both HumanEval (Chen, 2021) and MBPP (Austin et al., 2021). Additional implementation details are provided in Appendix H.

*Table 4.* Detailed results of the **off-policy data for coding.** Comparison of pass@1 scores on the HumanEval (HEval) and MBPP benchmarks.

| Method | DS-Coder (6.7B → 1.3B) | | | Qwen-Coder (7B → 1.5B ) | | |
|---|---|---|---|---|---|---|
| | HEval | MBPP | AVG | HEval | MBPP | AVG |
| $\mathcal{M}_T$ | 76.20 | 74.90 | 75.55 | 91.50 | 82.30 | 86.90 |
| $\mathcal{M}_S$ | 62.80 | 61.10 | 61.95 | 71.30 | 68.50 | 69.90 |
| SFT | 61.00 | 61.90 | 61.45 | 73.80 | 67.70 | 70.75 |
| KD | 65.20 | **64.00** | 64.60 | 77.40 | 67.50 | 72.45 |
| RKLD | 61.60 | 61.60 | 61.60 | 76.80 | 74.90 | 75.85 |
| JSD | 67.10 | 61.10 | 64.10 | 77.40 | 74.60 | 76.00 |
| **HPD** | **69.50** | 63.20 | **66.35** | **79.30** | **75.40** | **77.35** |

**Results and Analysis.** As shown in Table 4, HPD achieves the best average performance across both model families. While HPD does not always attain the highest score on every individual benchmark for DS-Coder, it demonstrates notably stable improvements. In contrast, KD and JSD exhibit larger variance across tasks and model families, indicating lower robustness. These results suggest that HPD provides a more balanced and reliable distillation objective.

## 5.4. On-policy Data for Reasoning

### 5.4.1. SETUP

*(1) Model and Dataset.* In this section, we follow the same model and dataset configuration as in Section 5.1, but switch to an online student rollout setting. *(2) Implementation Details.* We use a training batch size of 256, corresponding to 64 prompts with 4 repeated rollouts per prompt. We consider MiniLLM (Gu et al., 2023) and GKD (Agarwal et al., 2024), both of which fall under the OPD paradigm. The key difference is that MiniLLM emphasizes the role of reverse KL in improving distillation, while GKD centers on using on-policy student-generated sequences. Additional implementation details are provided in Appendix I.

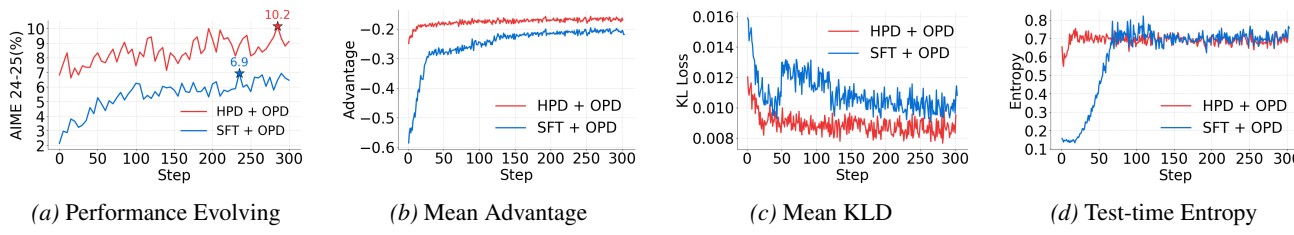

*(a)* Performance Evolving  *(b)* Mean Advantage  *(c)* Mean KLD  *(d)* Test-time Entropy

*Figure 2.* Training dynamics of OPD under different initializations.

*Table 5.* Detailed results of the **on-policy data for reasoning**. For AIME and AMC, the results are avg.@32. For the other benchmarks, the results are avg.@8. ∗ denotes that the performance is statistically significant (t-test with $p < 0.01$).

| | **Qwen 2.5 (7B → 1.5B)** | | | | | | | **LLaMA 3 (8B → 1B)** | | | | | | |
| Method | AIME24 | AIME25 | AMC | Math | Obly. | GPQA | Avg. | AIME24 | AIME25 | AMC | Math | Obly. | GPQA | Avg. |
|---|---|---|---|---|---|---|---|---|---|---|---|---|---|---|
| $\mathcal{M}_T$ | 28.13 | 27.19 | 71.72 | 87.48 | 58.50 | 43.43 | 52.74 | 14.27 | 18.02 | 55.23 | 77.78 | 47.74 | 36.23 | 41.55 |
| SFT | 2.81 | 6.04 | 28.83 | 55.25 | 24.87 | 19.02 | 22.80 | 0.83 | 1.04 | 17.34 | 33.30 | 12.52 | 18.24 | 13.88 |
| ↪ + OPD | 6.98 | 8.33 | 39.30 | 63.88 | 32.94 | 25.95 | 29.56 | 0.63 | 1.67 | 22.89 | 40.93 | 15.83 | 20.45 | 17.06 |
| **HPD** | 7.71 | 9.89 | 39.84 | 63.40 | 32.53 | **28.09**∗ | 30.24 | **1.25**∗ | 2.08 | 21.80 | 41.73 | 17.24 | **23.93** | 18.01 |
| ↪ + OPD | **10.63**∗ | **10.10**∗ | **43.98**∗ | **69.93**∗ | **38.59**∗ | 27.21 | **33.41** | 1.04 | **2.60**∗ | **28.68**∗ | **46.50**∗ | **19.93**∗ | 23.67 | **20.40** |

### 5.4.2. RESULTS AND ANALYSIS

**HPD exhibits stable optimization dynamics and effective policy alignment.** HPD consistently achieves higher task performance throughout training ( Figure 2a), which correlates with a less negative and more stable mean advantage estimate ( Figure 2b), indicating better alignment between the student and teacher policy. Meanwhile, HPD maintains a significantly lower KL divergence to the teacher ( Figure 2c), suggesting that HPD provides a more controlled and efficient correction of distribution mismatch than SFT. Importantly, HPD avoids excessive entropy collapse at test time ( Figure 2d), resulting in more consistent inference behavior that better matches the teacher's distribution.

**Effectiveness and Robustness of HPD.** HPD consistently outperforms SFT across different model families and benchmarks, demonstrating strong robustness and scalability. Even in the purely off-policy data setting, HPD alone achieves higher performance than the two-stage baseline that first applies SFT and then OPD. When combined with OPD, HPD further amplifies these gains, attaining the highest average scores and surpassing the corresponding baselines by a substantial margin. Notably, these improvements are consistent across both in-domain and out-of-domain evaluations (GPQA), indicating that HPD does not merely overfit to the teacher's behavior but instead transfers more generalizable decision signals.

## 6. Ablation Study

As formulated in Equation (14), HPD consists of two key components: (i) allowing the student to sample its own preferred actions, and (ii) reinforcing the expert token when

unreasonable actions are suppressed. To disentangle their individual contributions, we conduct ablation studies by removing each component in turn. Notably, HPD introduces no additional hyperparameters.

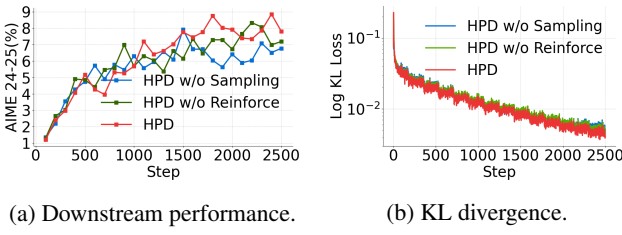

(a) Downstream performance.  (b) KL divergence.

*Figure 3.* Ablation study of HPD.

**Effectiveness of Student Sampling.** Without student sampling, the performance of the student model converges rapidly but quickly plateaus, failing to achieve sustained performance improvements. This behavior suggests that directly optimizing toward the teacher distribution limits exploration and leads to premature convergence. In contrast, enabling student sampling exposes the model to its own induced distribution, allowing it to explore diverse trajectories and continuously refine its policy, resulting in steady performance gains.

**Necessity of the Reinforce Operation.** As shown in Figure 3, removing the Reinforce operation results in slower KL loss reduction. By explicitly increasing the probability of expert tokens when unreasonable actions are suppressed, the Reinforce operation provides a more stable optimization signal, thereby accelerating alignment with the teacher distribution and resulting in consistent performance improvements.

*Table 6.* Efficiency comparison. The long-reasoning setup uses Qwen2.5-7B as the teacher and Qwen2.5-1.5B as the student, with batch size 32, sequence length 10,240, vocabulary size 152,064, and bfloat16 logits. Coding results use the 4096-token setting and greedy decoding.

| Metric | SFT | HPD | AKL | ABKD |
|---|---|---|---|---|
| Raw model-state memory | $\sim 22.35$ GiB | $\sim 35.39$ GiB | $\sim 35.39$ GiB | $\sim 35.39$ GiB |
| Extra logits/buffers | 0 GiB | $\sim 278$ GiB | $> 1.5$ TiB; $> 1.8$ TiB with cumulative buffers | $\sim 0.9$–$1.1$ TiB |
| Wall-clock relative to SFT | $1.0\times$ | $\sim 2.0$–$2.5\times$ | $\sim 3.5$–$5.0\times$ | $\sim 2.8$–$3.3\times$ |
| Coding train time | 3h20m | 7h30m | 27h | 14h |
| HumanEval / MBPP | 73.80 / 67.70 | **79.30** / **75.40** | 75.00 / **75.40** | 74.40 / 73.50 |

## 7. In-Depth Analysis

### 7.1. Computational Efficiency

A key advantage of HPD is that it uses teacher-distribution information without requiring full-vocabulary KL computation or top-$k$ distribution matching. Instead, HPD makes token-level decisions only on the expert token and a sampled student token.

As shown in Table 6, HPD keeps the model-state memory close to other white-box KD methods, but reduces the additional distribution-matching buffer from the TiB scale to about 278 GiB. Its wall-clock cost is also much closer to SFT than AKL (Wu et al., 2025) or ABKD (Wang et al., 2025). This efficiency comes from avoiding full-vocabulary or top-$k$ distribution matching and operating only on two token-level actions. In end-to-end accounting, HPD requires one teacher forward pass and obtains the student-sampled action during prefilling. By contrast, rollout-based OPD obtains student actions through decoding-time rollouts, making it substantially more expensive for long responses. In the coding setting, HPD obtains the best HumanEval score and matches the best MBPP score while requiring much less training time than AKL and ABKD. Overall, HPD provides a practical efficiency–performance trade-off for long-reasoning distillation.

### 7.2. Distillation from a Stronger Teacher

To further validate HPD under a larger teacher–student gap, we distill Qwen3-30B-A3B-Thinking-2507 into Qwen3-0.6B. While our main experiments use an 8K target length, we increase the training upper bound to 20K in this stronger-teacher study to further test the scaling behavior of HPD under our available computational budget. This length is substantially larger than the response lengths commonly used in prior KD settings, which are often 256, 512, or 1024 tokens (Wang et al., 2025; Agarwal et al., 2024). The corresponding training cost is discussed in Section 7.1.

As shown in Table 7, HPD consistently improves the Qwen3-0.6B student when distilled from the stronger Qwen3-30B-A3B-Thinking-2507 teacher. Under the 20K setting, HPD

*Table 7.* Stronger-teacher distillation from Qwen3-30B-A3B-Thinking-2507 to Qwen3-0.6B on AIME. Relative gains are computed against the original Qwen3-0.6B model under the same generation length.

| Gen Length | Bench | Original | HPD | Gain |
|---|---|---|---|---|
| 20K | AIME-24 | 11.25 | 12.19 | +8.36% |
| 20K | AIME-25 | 18.44 | 20.73 | +12.42% |
| 38K | AIME-24 | 11.35 | 15.78 | +39.03% |
| 38K | AIME-25 | 18.85 | 24.17 | +28.22% |

improves AIME-24 from 11.25 to 12.19 and AIME-25 from 18.44 to 20.73, corresponding to relative gains of 8.36% and 12.42%, respectively. When the evaluation cap is extended to 38K tokens, the gains become larger: AIME-24 improves from 11.35 to 15.78 and AIME-25 improves from 18.85 to 24.17, corresponding to relative gains of 39.03% and 28.22%. The distilled model also produces substantially longer reasoning traces on average, increasing from 16,719 to 29,112 tokens on AIME-24 and from 14,919 to 26,902 tokens on AIME-25.

## 8. Broader Impacts

Furthermore, we present a range of diverse applications for HPD, demonstrating its broad versatility and highlighting its potential for future use.

### 8.1. Additional Results for HPD + DPO

**Setup.** In preference alignment frameworks (Ouyang et al., 2022), model training is typically conducted in two stages: SFT and preference optimization via methods such as RLHF or DPO (Rafailov et al., 2023). Therefore, the initial phase plays an equally critical role. We select SFT and RKLD for comparison. We select the Qwen2.5-3B-Base model for analysis. Detailed settings are provided in Appendix J.

**Results.** As shown in Table 8, HPD lays the foundation for further optimization to build a strong small model. Since HPD aligns the student with the teacher distribution without inducing entropy collapse, it better facilitates subsequent alignment phases (Xiao, 2024). Our method achieves com-

*Table 8.* DPO stage results on Qwen2.5-3B (AlpacaEval2 / Arena-Hard) with different initialization methods.

| Method | AE-LC(%) | AE-WR(%) | Arena-WR(%) | Δ |
|---|---|---|---|---|
| $\mathcal{M}_T$ | 18.46 | 14.35 | 26.80 | – |
| SFT | 10.10 | 7.36 | 7.20 | – |
| ↪ + DPO | 10.42 | 9.27 | 10.40 | + 1.81 |
| RKLD | 11.13 | 9.35 | 13.80 | + 3.21 |
| ↪ + DPO | 15.45 | 16.78 | 21.80 | + 9.74 |
| **HPD** | 13.78 | 10.88 | 15.80 | + 5.27 |
| **↪ + DPO** | **17.68** | **17.65** | **25.10** | **+ 11.92** |

parable performance to the teacher model.

### 8.2. Iterative Self-Distillation Evolving

**Setup.** We alternately perform DPO training and self-distillation, where the teacher is the model obtained from the previous DPO stage. Throughout training, we retain the original SFT dataset, in contrast to approaches that replace SFT data with responses generated by the DPO-trained model (GLM, 2025). The training pipeline is shown in Figure 4.

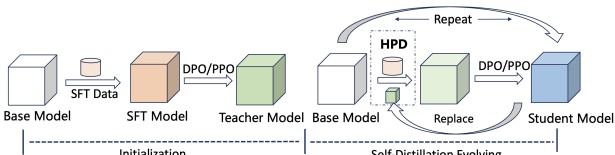

*Figure 4.* Self-distillation Evolving. Stage 1: SFT + DPO/PPO initialization. Stage 2: Iterative self-distillation with teacher model updates, while keeping the SFT data fixed.

*Table 9.* DPO stage results with iteratively updated teachers (AlpacaEval2 / Arena-Hard) on Qwen2.5-3B.

| Method | AE-LC(%) | AE-WR(%) | Arena-WR(%) | Δ |
|---|---|---|---|---|
| SFT | 10.10 | 7.36 | 7.20 | – |
| ↪ + DPO | 10.42 | 9.27 | 10.40 | + 1.81 |
| HPD-*iter1* | 11.77 | 9.76 | 12.50 | + 3.12 |
| ↪ + DPO | 13.67 | 13.83 | 16.30 | + 6.38 |
| HPD-*iter2* | 13.31 | 13.30 | 19.00 | + 6.98 |
| ↪ + DPO | 14.06 | 13.82 | 20.60 | + 7.94 |

**Results.** As shown in Table 9, HPD can transfer the teacher model's performance to the base model without performance dropping. By using DPO on the HPD model, the alignment performance continues to increase, resulting in performance scaling up. With the iteration epoch, the performance gain is limited. It further demonstrates strong-to-weak distillation as the start model is effective.

## 9. Related Work

**Off-policy Distillation** KD (Hinton et al., 2015) effectively compresses neural networks, allowing smaller student models to match the performance of larger teacher models. One popular direction of KD for LLMs is to directly harness teachers' generation as SFT data (Zhu et al., 2026). When the teacher model is accessible, it can utilize divergence loss to align the student's and teacher's distributions. Recent studies (Wang et al., 2025; Wen et al., 2023) have focused on finding the proper objectives to improve off-policy distillation. Wen et al. (2023) examined various f-divergences, including total variation distance and JSD, in auto-regressive LMs. Wu et al. (2025) also provided adaptive KL to balance their early-stage behaviors of FKLD and RKLD. Wang et al. (2025) formulates KD using an $\alpha$-$\beta$ divergence, enabling principled control over the probability mass allocation between the teacher and student distributions.

**On-policy Distillation** OPD encourages students to concentrate on a limited set of modes in the teacher's distribution. Both MiniLLM (Gu et al., 2023) and GKD (Agarwal et al., 2024) utilize RKLD or JSD to mitigate the student model's tendency to overestimate low-probability regions of the teacher. DistillLLM-2 (Ko et al., 2025) proposes a contrastive distillation formulation that simultaneously increases the likelihood of teacher-generated responses while decreasing that of student-generated responses, leveraging both off-policy and on-policy data.

## 10. Conclusion

We present a unified reweighted log-likelihood perspective on KD for LLMs, providing a principled understanding of how the direction of divergence, optimization strategy, and data regime jointly influence distillation behavior. Building on this view, we introduce HPD, a token-level distillation method that leverages a reverse-$k_1$ discrepancy signal together with prefilling-time student sampling to determine when to reinforce underestimated expert actions and when to suppress actions overestimated by the student. HPD retains the efficiency of SFT-style one-hot supervision while avoiding the overhead of full-vocabulary or top-k distribution matching and costly full-sequence student rollouts. Future work may explore applying HPD at earlier stages of model development, such as mid-training or even pre-training, to further evaluate its potential.

### Acknowledgement

This work was partially funded by the National Natural Science Foundation of China (62476168), SJTU School of Electronic Information and Electrical Engineering – ByteDance LLM Joint Laboratory, and the Young Elite Scientists Sponsorship Program by CAST (2023QNRC001).

## Impact Statement

This paper presents work whose goal is to advance the field of machine learning, specifically the efficiency and efficacy of LLMs. There are many potential societal consequences of our work, none of which we feel must be specifically highlighted here.

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

# Appendix

## A. Positioning of Our Work

Our work focuses on distillation over off-policy data, augmented with a lightweight approximation of on-policy next-token sampling, which avoids full-sequence rollouts. While we acknowledge that full rollouts—such as those used in OPD (Lu & Lab, 2025) or reinforcement learning (RL) (Schulman et al., 2017; Shao et al., 2024)—are necessary to further enhance distilled model performance, our method is designed to provide a more robust and efficient initialization for subsequent training, and in some cases achieves strong performance without requiring these rollout-based methods.

## B. Limitation

Our work is limited to settings where the teacher and student models share the same tokenizer, which is a common assumption in white-box KD (Jiao et al., 2020; Wang et al., 2025; Ko et al., 2024; 2025). Due to the intractability of computing the full KL divergence, we instead adopt the corresponding estimators to provide distillation signals on the token level.

### B.1. Gradient Analysis with $K_1$ reward

In this section, we investigate the role of $K_1$ as a reward signal in gradient analysis. Let $\tau = (s_1, a_1, s_2, a_2, \ldots, s_T, a_T)$ denote a trajectory sampled from the student policy $q_\theta$, where each state $s_t$ corresponds to the prefix $s_t = \mathbf{a}_{<t} = (a_1, \ldots, a_{t-1})$, and $a_t \in \mathcal{V}$ is the token sampled at step $t$ from a finite vocabulary $\mathcal{V}$. Let $\widehat{\mathrm{KL}}$ denotes the Monte Carlo (MC) estimate of the KL divergence.

$$\nabla_\theta \mathbb{E}_{\tau \sim q_\theta}[\widehat{\mathrm{KL}}] = \nabla_\theta \sum_\tau q_\theta(\tau)\,\widehat{\mathrm{KL}}(\tau) \tag{16}$$

$$= \sum_\tau \left(\nabla_\theta \widehat{\mathrm{KL}}(\tau)\right) q_\theta(\tau) + \sum_\tau \widehat{\mathrm{KL}}(\tau)\,\nabla_\theta q_\theta(\tau) \tag{17}$$

$$= \underbrace{\mathbb{E}_{\tau \sim q_\theta}\left[\sum_{t=1}^T \nabla_\theta \widehat{\mathrm{KL}}_t\right]}_{\text{path-wise derivative}} + \underbrace{\mathbb{E}_{\tau \sim q_\theta}\left[\left(\sum_{t=1}^T \widehat{\mathrm{KL}}_t\right)\nabla_\theta \log q_\theta(\tau)\right]}_{\text{score function derivative}}. \tag{18}$$

Here, the path-wise derivative corresponds to backpropagating directly through the estimator in the loss, whereas the score function derivative corresponds to treating the estimator as a reward signal. We adopt the $K_1$ estimator with $N = 1$ in Equation (8) defined as

$$K_1 \triangleq \sum_{t=1}^T \log \frac{q_\theta(a_t \mid s_t)}{p(a_t \mid s_t)}, \quad a_t \sim q_\theta(\cdot \mid s_t),$$

whose path-wise derivative is $\mathbb{E}_{\tau \sim q_\theta}\left[\nabla_\theta \sum_{t=1}^T \mathrm{K}_{1t}\right] = 0$, and whose score function derivative is

$$\mathbb{E}_{\tau \sim q_\theta}\left[\left(\sum_{t=1}^T \mathrm{K}_{1t}\right) \cdot \nabla_\theta \log q_\theta(\tau)\right] = \mathbb{E}_{\tau \sim q_\theta}\left[\log \frac{q_\theta(\tau)}{p(\tau)} \cdot \nabla_\theta \log q_\theta(\tau)\right]. \tag{19}$$

Thus, incorporating the $K_1$ estimator as a reward provides an unbiased estimate of the gradient for the KLD objective, whereas using $K_1$ directly in the loss does not.

## C. Gradient of Jensen-Shannon Divergence

Let $p(a_t \mid s_t)$ and $q_\theta(a_t \mid s_t)$ be the teacher and student distributions over tokens $a_t \in \mathcal{V}$ given state $s_t$. The Jensen-Shannon divergence (JSD) is defined as:

$$\mathrm{JSD}(p \parallel q_\theta) = \frac{1}{2}\mathbb{D}_{\mathrm{KL}}(p \parallel M) + \frac{1}{2}\mathbb{D}_{\mathrm{KL}}(q_\theta \parallel M), \quad M(a_t \mid s_t) = \frac{p(a_t \mid s_t) + q_\theta(a_t \mid s_t)}{2}.$$

For discrete distributions, the JSD can be written as:

$$\mathrm{JSD}(p \parallel q_\theta) = \frac{1}{2}\sum_{a_t} p(a_t \mid s_t)\log\frac{p(a_t \mid s_t)}{M(a_t \mid s_t)} + \frac{1}{2}\sum_{a_t} q_\theta(a_t \mid s_t)\log\frac{q_\theta(a_t \mid s_t)}{M(a_t \mid s_t)}.$$

The gradient w.r.t. $\theta$ only depends on the second term, since the first term does not involve $\theta$:

$$\nabla_\theta\mathrm{JSD}(p \parallel q_\theta) = \frac{1}{2}\sum_{a_t}\nabla_\theta q_\theta(a_t \mid s_t)\log\frac{q_\theta(a_t \mid s_t)}{M(a_t \mid s_t)}$$

$$= \sum_{a_t} q_\theta(a_t \mid s_t)\nabla_\theta\log q_\theta(a_t \mid s_t) \cdot \underbrace{\frac{1}{2}\log\frac{q_\theta(a_t \mid s_t)}{M(a_t \mid s_t)}}_{w_{\mathrm{JSD}}(a_t \mid s_t)}$$

$$= \mathbb{E}_{a_t \sim q_\theta(\cdot \mid s_t)}\Big[\nabla_\theta\log q_\theta(a_t \mid s_t) \cdot w_{\mathrm{JSD}}(a_t \mid s_t)\Big].$$

Here, $w_{\mathrm{JSD}}(a_t \mid s_t) = \frac{1}{2}\log\frac{q_\theta(a_t \mid s_t)}{M(a_t \mid s_t)}$ can be interpreted as a token-level weight, analogous to the reward weights in SFT, FKLD, and RKLD.

## D. Gradient contribution

The resulting gradient of Equation (9) on a sampled token $a_t$ can be expressed as:

$$-\frac{\partial\mathcal{L}(\theta)}{\partial z_v} \propto \begin{cases} \hat{w}_t \cdot q_v \cdot (1 - q_v), & \text{if } v = a_t, \\ -\hat{w}_t \cdot q_{a_t} \cdot q_v, & \text{if } v \neq a_t, \end{cases} \tag{20}$$

where $z_v$ denotes the logit associated with token $v$ and $\hat{w}_t$ abbreviates $w(a_t \mid s_t)$.

*Proof.* To compute the gradient with respect to $z_v$, we apply the chain rule:

$$\frac{\partial\mathcal{L}_t}{\partial z_v} = \frac{\partial\mathcal{L}_t}{\partial q_{a_t}} \cdot \frac{\partial q_{a_t}}{\partial z_v}. \tag{21}$$

Since

$$\frac{\partial\mathcal{L}_t}{\partial q_{a_t}} = -\hat{w}_t\frac{1}{q_{a_t}}, \tag{22}$$

It remains to compute the derivative of the softmax function. We now derive $\frac{\partial q_{a_t}}{\partial z_v}$ by considering two cases.

**Case 1:** $v = a_t$. Using the quotient rule, we obtain

$$\frac{\partial q_{a_t}}{\partial z_{a_t}} = \frac{\partial}{\partial z_{a_t}}\left(\frac{e^{z_{a_t}}}{\sum_k e^{z_k}}\right) = \frac{e^{z_{a_t}}\sum_k e^{z_k} - e^{z_{a_t}}e^{z_{a_t}}}{\left(\sum_k e^{z_k}\right)^2}. \tag{23}$$

Simplifying yields

$$\frac{\partial q_{a_t}}{\partial z_{a_t}} = q_{a_t}(1 - q_{a_t}). \tag{24}$$

Substituting back, we have

$$\frac{\partial\mathcal{L}_t}{\partial z_{a_t}} = -\hat{w}_t(1 - q_{a_t}), \tag{25}$$

and therefore

$$-\frac{\partial\mathcal{L}_t}{\partial z_{a_t}} \propto \hat{w}_t\,q_{a_t}(1 - q_{a_t}). \tag{26}$$

**Case 2:** $v \neq a_t$. Similarly,

$$\frac{\partial q_{a_t}}{\partial z_v} = \frac{\partial}{\partial z_v}\left(\frac{e^{z_{a_t}}}{\sum_k e^{z_k}}\right) = -\frac{e^{z_{a_t}} e^{z_v}}{\left(\sum_k e^{z_k}\right)^2}. \tag{27}$$

Rewriting in terms of softmax probabilities gives

$$\frac{\partial q_{a_t}}{\partial z_v} = -q_{a_t} q_v. \tag{28}$$

Thus,

$$\frac{\partial \mathcal{L}_t}{\partial z_v} = \hat{w}_t q_v, \tag{29}$$

and equivalently,

$$-\frac{\partial \mathcal{L}_t}{\partial z_v} \propto -\hat{w}_t\, q_{a_t} q_v, \qquad v \neq a_t. \tag{30}$$

Combining both cases yields the gradient expression in Eq. (20).

$\square$

# E. Baselines

Here, we present several baselines as follows:

- **SFT** is supervised fine-tuning of the student model using ground-truth on the fixed dataset (using predefined input-output pairs)

- **KD** (Hinton et al., 2015) trains the student distribution to mimic the teacher distribution on the fixed dataset using FKLD.

- **SeqKD** (Kim & Rush, 2016) maximizes the likelihood of high probability sequences generated by the teacher, and can be viewed as SFT on teacher-generated outputs.

- **RKLD.** MiniLLM (Gu et al., 2023) performs distillation on student-generated data using an on-policy method to minimize the RKLD between the teacher and student distributions. In contrast, we extend this objective to the off-policy setting in Sections 5.1 to 5.3, and additionally present on-policy experiments in Section 5.4.

- **JSD.** GKD (Agarwal et al., 2024) uses the generalized Jensen-Shannon divergence $\left(\mathbb{D}_{\text{JSD}(\beta)}(p\|q_\theta) = \beta \mathbb{D}(p\|\beta p + (1-\beta)q_\theta) + (1-\beta)\mathbb{D}(q_\theta\|\beta p + (1-\beta)q_\theta)\right)$, training on a mixture of datasets, either teacher-generated or ground-truth, and on-policy student-generated sequences. We perform GKD on ground-truth data in Section 5.1, on teacher-generated data in Sections 5.2 and 5.3, and in an on-policy setting in Section 5.4.

# F. Off-policy Data for Reasoning

## F.1. Learning Rate

We fine-tune the corresponding base model for 10 epochs using the 40k data described in Section 5.1. The base model and initial learning rate are listed in Table 10. We employ a *cosine decay* learning rate scheduler that gradually decreases the learning rate to one-tenth of its initial value. The maximum context length is 10240 tokens, and the batch size is 256.

*Table 10.* Learning Rate Configuration

| Model | Initial Learning Rate | min_lr_ratio |
|---|---|---|
| Qwen2.5-1.5B | $1 \times 10^{-4}$ | 0.1 |
| Qwen2.5-3B | $8 \times 10^{-5}$ | 0.1 |
| Llama3.2-1B | $8 \times 10^{-5}$ | 0.1 |
| Llama3.2-3B | $5 \times 10^{-5}$ | 0.1 |

## F.2. Teacher models

We select Qwen-2.5-7B and Llama-3.1-8B models to perform SFT learning on the OpenR1-Math-8192 dataset (Face, 2025), and use DAPO (Yu et al., 2025) with a clip-higher value of 0.28, a stable variant of GPPO. The RL training uses the DAPO-MATH-17k dataset, with detailed training configurations of verl framwork (Sheng et al., 2024) provided as follows:

*Table 11.* RL experiment configuration

| Config | RL |
|---|---|
| lr | 1e-6 |
| kl_coef | 0.0 |
| max_prompt_length | 2k |
| max_response_length | 10k 0k |
| overlong_buffer.len | 2k |
| train_batch_size | 256 |
| ppo_mini_batch_size | 32 |
| clip_ratio_low | 0.2 |
| clip_ratio_high | 0.28 |
| temperature | 1.0 |
| rollout.n | 8 |
| total_training_steps | 100 |

Since RL primarily fine-tunes small subnetworks within LLMs (Mukherjee et al., 2025) and induces relatively small KL deviations (Shenfeld et al., 2025), it can further improve the teacher model's performance.

## F.3. Evaluation Benchmark.

*(i) In-domain tasks:* AIME24, AIME25, AMC, MATH-500 (Hendrycks et al., 2021), and OlympidBench (He et al., 2024).

*(ii) Out-of-domain tasks:* GPQA (Rein et al., 2024)

For all evaluations, we generate responses with a maximum sequence length of 10,240, using top-$p$ sampling of 0.95 and a temperature of 0.7.

# G. Off-policy Data for Personalization

## G.1. Learning Rate

We fine-tune the corresponding base model for 10 epochs using the UltraFeedback dataset (Cui et al., 2023), consisting of 4,096-token teacher-generated data as described in Section 5.2. We use a learning rate of $5 \times 10^{-6}$, a maximum context length of 4,096, and a batch size of 256.

## G.2. Evaluation Benchmark.

We evaluate our models on three alignment benchmarks: MT-Bench (Zheng et al., 2024a), AlpacaEval (Dubois et al., 2024), and Arena-Hard (Li et al., 2024). We use Qwen3-30B-A3-Instruct-2507 (Yang et al., 2025) as the judge model to provide alignment evaluation.

For all evaluations, we generate responses with a maximum sequence length of 4,096, using top-$p$ sampling with $p = 0.95$ and a temperature of 0.7. For LLM-as-a-Judge evaluations, we set the temperature to 0.0.

# H. Off-policy Data for Coding

## H.1. Learning Rate

We fine-tune the corresponding base model for 10 epochs on the WizardCode dataset (Luo et al., 2024), containing 4,096-token teacher-generated data ( Section 5.3). Both Qwen and DeepSeek models use a learning rate of $5 \times 10^{-6}$, maximum

context length of 4,096, and batch size of 256.

## I. On-policy Data for Reasoning

Both Qwen and LLaMA use a learning rate of $3 \times 10^{-6}$ with on-policy student rollouts, and evaluation is conducted as described in Section F.3. We use a training batch size of 256, corresponding to 64 prompts with 4 repeated rollouts per prompt, following the same setting as in Lu & Lab (2025).

## J. Broder Impact

In Section 8, we select Qwen2.5-7B-Base model as the teacher. We first SFT all the base models and further align the model using the DPO algorithm (Rafailov et al., 2023) implemented with LLama-Factory (Zheng et al., 2024b). The detailed training configurations for DPO are summarized in Table 12.

*Table 12.* DPO experiment configuration

| Method | Train batch size | $\beta$ | Learning rate | Train epochs | Cutoff_len |
|--------|------------------|---------|---------------|--------------|------------|
| DPO | 64 | 0.01 | 5e-7 | 1 | 4k |

### J.1. System Prompt

For the reasoning task, we use the system prompt as follows: `Please reason step by step, and put your final answer within \boxed{}`. For the alignment task and code task, we use the system prompt as follows: `You are a helpful assistant.`

### J.2. Data Decontamination

We conduct a decontamination analysis between the training data and evaluation benchmarks. We first apply both light and aggressive normalization to detect exact matches. We then construct 6-gram candidates and apply `SequenceMatcher` to identify near-duplicate variants. AIME-24, AIME-25, AMC, GPQA, and Minerva have zero detected overlap under this procedure. Only MATH-500 and OlympiadBench contain flagged rows, as summarized in Table 13. These flagged examples are reported for transparency and manually audited before final evaluation.

*Table 13.* Detected overlap candidates from the decontamination analysis.

| Dataset | High-confidence rows | Possible variant rows |
|---------|----------------------|------------------------|
| MATH-500 | 16 | 16 |
| OlympiadBench | 96 | 0 |

### J.3. Confidence Intervals

For alignment evaluation, we report Arena-Hard scores with $95\%$ confidence intervals in Table 14. Coding tasks are evaluated with greedy decoding.

## K. Case Study

Given prefixes sampled from UltraFeedback, we use both the teacher and distilled student models to generate responses with greedy decoding. Representative cases are shown below. We prompt the models with: `"Could you recommend a hotel in Istanbul with a rooftop pool and bar for relaxation?"`

> **Teacher: Qwen2.5-7B-Instruct**
>
> Certainly! In Istanbul, one of the hotels that offers a rooftop pool and bar where you can relax while enjoying panoramic views is the **Park Hyatt Istanbul**.

*Table 14.* Arena-Hard scores with 95% confidence intervals.

| Model | Arena-Hard | 95% CI |
|-------|-----------|--------|
| Base | 7.47 | $(-1.0,\ 1.1)$ |
| SFT | 18.10 | $(-1.5,\ 1.5)$ |
| SeqKD | 15.40 | $(-1.2,\ 1.2)$ |
| RKLD | 17.80 | $(-1.1,\ 1.0)$ |
| JSD | 20.20 | $(-1.5,\ 1.6)$ |
| HPD | **21.80** | $(-0.9,\ 1.2)$ |

---

**Park Hyatt Istanbul:**

- **Location:** The hotel is located in the heart of Istanbul, near Tophane Square and the Bosphorus.

- **Rooftop Facilities:**

  – **Pool:** The hotel features an adults-only infinity pool on its rooftop, offering stunning views of the city and the Bosphorus.
  – **Bar:** The rooftop bar, named "Tophane Bar," provides a perfect spot to unwind with cocktails while taking in the breathtaking views.

- **Additional Amenities:**

  – Spa and wellness center
  – Multiple dining options including Turkish cuisine, Italian restaurant, and a gourmet food hall
  – Fitness center
  – Concierge services

This hotel is highly recommended for its luxurious accommodations, exceptional service, and the unique feature of its rooftop pool and bar. It's particularly popular among travelers seeking a high-end experience in a prime location. If you have any specific preferences or requirements, feel free to let me know, and I can provide more tailored recommendations!

---

Student: Qwen2.5-1.5B-SFT

Certainly! One highly recommended hotel in Istanbul that offers a rooftop pool and bar is the **Park Hyatt Istanbul**.

**Park Hyatt Istanbul:**

- **Location:** Istiklal Avenue, Sultanahmet, Istanbul, Turkey

- **Rooftop Pool & Bar:** The hotel features an expansive rooftop garden with a stunning infinity pool, offering breathtaking views of the city skyline. It also includes a vibrant bar where you can enjoy refreshing drinks while taking in the atmosphere.

- **Other Amenities:**

  – **Relaxation Spaces:** The rooftop area provides a perfect spot for unwinding after a day of exploration.
  – **Dining Options:** Besides the bar, there are dining options available on-site that offer Turkish and international cuisine.
  – **Accessibility:** The hotel is conveniently located near major attractions such as Hagia Sophia, Topkapi Palace, and the Blue Mosque.

This hotel is ideal for those looking to experience the best of Istanbul's culture and ambiance while enjoying a tranquil and relaxing environment.

---

### Student: Qwen2.5-1.5B-KD

Certainly! One highly recommended hotel in Istanbul that offers a rooftop pool and bar is the **Park Hyatt Istanbul**.

**Park Hyatt Istanbul:**

- **Location:** Located in the heart of Istanbul's Grand Bazaar district, this luxurious hotel provides easy access to the city's vibrant attractions.

- **Rooftop Pool & Bar:** The hotel features an expansive rooftop pool with stunning views of the city skyline. It also includes a charming bar where you can enjoy refreshments while taking in the scenic views.

- **Accommodations:** The hotel offers modern, elegantly designed rooms and suites, providing ample space for relaxation and comfort.

- **Additional Amenities:** Other amenities include a spa, restaurant, and a 24/7 concierge service to help with any needs or requests.

This hotel is perfect for those looking to relax and unwind in a cozy yet luxurious setting after exploring the bustling city life. Enjoy your stay! If you need more information or have any other questions, feel free to ask!

---

### Student: Qwen2.5-1.5B-HPD

Certainly! One highly recommended hotel in Istanbul with a rooftop pool and bar is the **Park Hyatt Istanbul**. This luxurious hotel offers stunning views of the Bosphorus Strait from its rooftop pool area, which provides guests with a perfect spot to relax and enjoy the city skyline.

**Key Features:**

- **Rooftop Pool:** The hotel's rooftop pool offers breathtaking views and is a great place to unwind after a day exploring Istanbul.

- **Bar:** The bar on the rooftop also serves as a vibrant social hub where you can enjoy drinks and snacks while taking in the panoramic views.

- **Luxury Amenities:** The hotel itself is known for its high-end amenities, including 24-hour room service, a spa, and multiple dining options.

**Location:**
The Park Hyatt Istanbul is conveniently located near key attractions such as the Hagia Sophia, Topkapi Palace, and the Grand Bazaar, making it an ideal base for both sightseeing and relaxation.
If you're looking for more specific details or have any other questions about your trip to Istanbul, feel free to ask!

---

### Student: Qwen2.5-1.5B-JSD

Certainly! One highly recommended hotel in Istanbul with an excellent rooftop pool and bar is the **Four Seasons Hotel Istanbul**.

**Four Seasons Hotel Istanbul:**

- **Location:** Located on the Bosphorus Strait, this luxurious hotel offers stunning views of both the city and the water.

- **Rooftop Pool & Bar:**
  - The rooftop pool is one of the most popular features, offering panoramic views of the Bosphorus and the city skyline.
  - It's perfect for enjoying a relaxing swim or sunbathing while taking in the beautiful surroundings.
  - The bar provides a vibrant atmosphere to enjoy cocktails, snacks, and light meals.

- **Other Amenities:**
  - The hotel also includes a spa, fitness center, and multiple dining options.
  - It's conveniently located near major attractions like Hagia Sophia and Topkapi Palace.

This hotel combines luxury with a fantastic location and amenities, making it an ideal choice for those looking to relax and enjoy their stay in Istanbul.

---

Student: Qwen2.5-1.5B-RKLD

Certainly! One highly recommended hotel in Istanbul with a rooftop pool and bar is the **Park Hyatt Istanbul**.

**Park Hyatt Istanbul:**

- **Location:** Located in the heart of Istanbul's Grand Bazaar area, this luxurious hotel offers easy access to historical sites and bustling city life.

- **Rooftop Pool & Bar:**
  - The hotel features an expansive rooftop pool where guests can relax and enjoy stunning views of the city skyline.
  - There's also a vibrant rooftop bar offering a variety of cocktails and refreshments, perfect for socializing or unwinding after a day exploring Istanbul.

**Other Options:**

1. **Istiklal Avenue Hotel:**

   - Known for its elegant ambiance and proximity to major attractions like the Blue Mosque and Topkapi Palace.
   - It doesn't have a rooftop pool but does offer great views from its rooms and common areas.

2. **Grand Hyatt Istanbul:**

   - Situated near the Bosphorus Strait, it provides panoramic sea views from its rooftop terrace.
   - While not exclusively a pool hotel, they do have a large outdoor space for relaxation.

3. **Marmara Hotel:**

   - Located on the Asian side of Istanbul, it offers a blend of modern amenities and traditional Turkish charm.
   - They don't have a specific rooftop pool, but there are plenty of other ways to enjoy the outdoors, such as their rooftop garden.

Each of these hotels offers unique experiences, so choose based on your preferences and what you hope to get out of your stay in Istanbul. Enjoy your trip!

---

**Analysis.** As shown above, the distillation methods using SFT, KD, HPD, and RKLD all recommend the Park Hyatt Istanbul. In contrast, JSD provides a different recommendation. Notably, RKLD, in addition to recommending a single hotel, also lists alternative options. Overall, HPD tends to provide more detailed information, and its responses include thoughtful greetings tailored to different user needs, similar to those generated by the teacher model.

