# OpenReview forum: "Hybrid Policy Distillation for LLMs"
_ICML.cc/2026/Conference — ICML 2026 regular_

### Official Review · Reviewer_FeSF · 2026-02-26

**Soundness:** 2
**Presentation:** 3
**Significance:** 2
**Originality:** 2
**Overall Recommendation:** 3
**Confidence:** 4

**Summary:**

This paper focuses on knowledge distillation of LLMs. Specifically, the authors propose a reweighted log-likelihood method that combines the forward KL and reverse KL via lightweight on-policy sampling.

**Compliance With Llm Reviewing Policy:**

Affirmed.

**Key Questions For Authors:**

1 In Line 84, the paper states “Experiments on math reasoning … pipelines with less cost”. And in Line 28 the paper states “demonstrating improved optimization stability, computational efficiency”. But there are no clear experimental reports in experiment section.

2 the author provide a unified view of Distillation and combine the FKL and RKL. What is the main motivation for proposing this method? What difficulties of exiting distillation does it aim to solve?

**Limitations:**

The author write the future work in the end of Conclusion section.

**Strengths And Weaknesses:**

Strengths：

1 This paper is well-structured and logically organized, and the proposed method is easy to follow.

2 The authors conduct various experiments, and the results show considerable improvements.

Weaknesses：

1 The novelty of this paper appears limited. Actually, there are some existing methods have explored the similar ideas: combined FKL and RKL adaptably, such as AKL[1]. In addition, the experiment section does not include any reweighting baselines.

[1] Rethinking Kullback-Leibler Divergence in Knowledge Distillation for Large Language Models. COLING 2025

2 In Section 4.2, the authors define the expert token as offline ground truth that approximated teacher distribution, but in practice, it is often unavailable to get the teacher distribution or the training data of teacher. Could the authors clarify how these expert tokens are obtained?

---

> ### Author Rebuttal · Authors · 2026-03-30
>
> > 【W1】The novelty of this paper appears limited. Actually, some existing methods have explored similar ideas: combined FKL and RKL adaptively,
>
> (1) **Novelty:**  Our work introduces several novel contributions compared to existing methods. First, we achieve the **efficiency of one-hot targets** similar to SFT while outperforming other baselines that rely on full or top-k distributions. Another key innovation is our **mask mechanism**, which utilizes a reverse-KL estimator to determine when to apply FKLD or RKLD, in contrast to methods that directly fuse them. Additionally, we employ **lightweight on-policy sampling** to effectively balance exploration and exploitation, eliminating the need for full-sequence rollouts.
>
> (2) We give the following baselines to demonstrate HPD's better performance.
>
> **Baseline Efficiency**: We compare computational resources as follows: using Qwen2.5-7B as the teacher and Qwen2.5-1.5B as the student, with a batch size of 32, a sequence length of 10240, and a vocabulary size of 152064. The memory for a single tensor at bfloat16 precision is 92.81 GiB.
>
> | Metric | SFT | HPD | AKL| ABKD |
> | ---------------------------- | ---------- | ----------- | ---------- | ---------- |
> | Raw model-state memory | ~22.35 GiB | ~35.39GiB | ~35.39GiB  | ~35.39 GiB |
> | Extra logits | ~92.8GiB | ~370.8 GB | ~2.2TiB | ~2.5TiB |
> | Wall-clock (compared to SFT) | 1.0x | ~2.3x -2.7x | ~3.5x - 5x | ~2.8x-3.3x |
>
> **Baseline Performance:**  For coding tasks, we use 4096 tokens for distillation. Performance results are as follows:
>
> | Metric | SFT | ABKD | AKL | HPD|
> | ---------- | ------- | -------- | -------- | --------- |
> | HumanEval | 73.80 | 74.40  | 75.00 | **79.30**|
> | MBPP  | 67.70 | 73.50 | 75.40 | **75.40**|
> | Train Time |3h20|14h|27h|7h30|
>
> **Conclusion: HPD offers significant efficiency gains while delivering superior performance compared to other methods.**
>
> > 【W2】It is often difficult to get the teacher distribution or the training data of the teacher. Could the authors clarify how these expert tokens are obtained?
>
> (1) **We focus on white-box offline distillation**, where the teacher model and offline dataset are accessible, either from the open-source community or through in-house distillation.
>
> (2) The training data of the teacher is not required; instead, we leverage the teacher model’s distribution over the offline dataset.
>
> > 【Q1】No clear experimental reports about optimization stability and efficiency in the experiment section.
>
> **(1) Computational Efficiency:** See W1; we will include this in the paper and make an explicit description.
>
> **(2) Optimization Stability:** We highlight four aspects to demonstrate optimization stability:
>
> 1. **Overfitting:** Figure 1(a) shows our method reduces overfitting compared to SFT. Figure 1(d) demonstrates that, after HPD optimization, the student model better aligns with the teacher's distribution at inference.
> 2. **Performance:** As the gap between student and teacher narrows, the evaluation performance improves, as shown in Figures 1(b) and 1(c).
> 3. **Gradient Norm Analysis:** HPD achieves better optimization stability than SFT, as evidenced by a lower final gradient norm (~0.2 vs. ~0.4) and reduced variance throughout training. The gradient dynamics can be found in this anonymous link: https://anonymous.4open.science/r/HPD-E273/Grad_norm.png
> 4. **Robustness:** Results from Math, Coding, and Alignment tasks show that HPD outperforms other methods.
>
>
> > 【Q2】The author provides a unified view of distillation and combines the FKL and RKL. What is the main motivation for proposing this method? What difficulties of exiting distillation does it aim to solve?
>
> (1) We begin by providing a unified view of the distillation schema, followed by a practical solution in the form of our designed algorithm, HPD.
>
> (2) The primary motivation for proposing a unified distillation framework is to integrate various distillation techniques into a cohesive system, enhancing the understanding and design of new algorithms while better capturing the interactions between off-policy, on-policy, forward, and reverse signals and data.
>
> (3) **HPD addresses the following:**
>
> 1. **Balancing FKLD and RKLD well:**
>     HPD uses a **masking mechanism** to focus on the student's underestimated expert tokens and suppress overestimated ones, avoiding gradient contradictions seen in methods that directly combine both terms.
> 2. **Balancing Exploration and Exploitation:**
>     HPD combines offline learning with lightweight sampling in the pre-filling phase, achieving a balance between **exploration** and **exploitation**. Unlike on-policy methods, which depend heavily on student rollouts.
> 3. **Efficient Distillation:**
>     HPD merges the computational efficiency of **one-hot targets** with the teacher model's forward pass, while other methods rely on top-k or full distribution matching.
>
> ---
> **We hope we have addressed your concerns, and hope you can provide a better judgment of our work.**

---

> > ### Author Rebuttal · Reviewer_FeSF · 2026-04-05
> >
> > Thanks for your response, I will raise my score.

---

> > > ### Author Response · Authors · 2026-04-08
> > >
> > > **Thank you for recognizing our efforts and for being willing to raise your score. We are glad that our explanations have addressed your concerns.**

---

### Official Review · Reviewer_JzR7 · 2026-03-07

**Soundness:** 3
**Presentation:** 3
**Significance:** 3
**Originality:** 3
**Overall Recommendation:** 4
**Confidence:** 4

**Summary:**

This study studies knowledge distillation (KD) for compressing large language models (LLMs) by unifying divergence direction, optimization strategy, and data regime at the token level. The authors propose Hybrid Policy Distillation (HPD), which integrates forward and reverse KL divergences while combining off-policy data with lightweight on-policy sampling. Experiments demonstrate that HPD improves optimization stability, computational efficiency, and achieves performance gains over multiple baselines.

**Compliance With Llm Reviewing Policy:**

Affirmed.

**Final Justification:**

This paper presents a systematic and well-motivated study on knowledge distillation for LLMs. The proposed HPD method demonstrates consistent empirical improvements across multiple model families and tasks.
The authors' rebuttal has addressed most of my concerns. In particular, they clarified the rationale behind using offline trajectories for student sampling and how HPD reconciles the apparent mismatch with on-policy data, which I found convincing. Additional experimental results provided in the rebuttal further support the design choices of the method. Overall, the rebuttal has strengthened my confidence in the paper, and I have raised my score accordingly.

**Key Questions For Authors:**

1. Please clarify the rationale for using offline trajectories for student sampling in HPD instead of student rollouts.
2. Can the authors comment on the applicability of HPD to more advanced teacher models (e.g., Qwen 3/3.5)?
3. How does HPD reconcile offline sampling with on-policy data in Section 5.4?

**Limitations:**

Yes.

**Strengths And Weaknesses:**

Strengths:

1. The paper systematically investigates the effects of divergence direction, optimization strategy, and data regime on KD for LLMs, offering a clear unified perspective.
2. HPD combines forward and reverse KL and leverages both offline and lightweight on-policy data, providing a principled approach that balances mode coverage and mode-seeking behaviors.
3. HPD consistently outperforms standard KD baselines across multiple model families (Qwen 2.5 and LLaMA 3) and tasks (math, dialogue, code), demonstrating robust improvements.

Weaknesses:

1. The student is sampled based on offline trajectories rather than on its own rollouts, which is counter-intuitive. Traditionally, on-policy distillation would encourage the student to learn from its own trajectories using teacher guidance. The rationale for using offline trajectories in HPD is unclear and may limit generalization.
2. Experiments use Qwen 2.5 and LLaMA 3, whose performance on difficult reasoning benchmarks (e.g., AIME24/25) is unsatisfactory. The evaluation may not fully reflect HPD’s potential on more advanced LLMs such as Qwen 3 or Qwen 3.5, especially in challenging reasoning tasks.
3. In Section 5.4, HPD relies on offline trajectories for sampling, yet the on-policy data setting involves student-generated rollouts. It is unclear how HPD reconciles this mismatch in practice, and whether the method can fully leverage on-policy feedback.

---

> ### Author Rebuttal · Authors · 2026-03-30
>
> > 【W1 & Q1】 Please clarify the rationale for using offline trajectories for student sampling in HPD instead of student rollouts.
>
> Many distillation methods, such as SFT, AKL, and ABKD, rely on offline trajectories. Our approach aligns with these methods, but **extends the use of offline trajectories to predict the next token.** This strategy ensures that the student not only imitates the offline data but also learns to recognize its own mistakes, achieving a better balance between **efficient exploration** and **exploitation**.
>
> When compared to SFT using only offline trajectories, HPD performs better and improves further with additional OPD.
>
>
> This approach offers several advantages:
>
> 1. **Avoids the heavy reliance on rollouts** in on-policy data.
> 2. **Eliminates the training-inference mismatch** by ensuring that the training data closely matches the teacher’s distribution.
> 3. **Stabilizes training**, reducing the risk of unstable gradients and poor convergence that often arise from using student-generated data during training.
> 4. **Improves generalization** by aligning the student’s output with the teacher's distribution, compared to SFT’s reliance on the distillation dataset. (See Table 2 and Table 5 for the GPQA metrics.)
> 5. **Optimizes the exploration-exploitation tradeoff**, enabling the student to efficiently explore while still focusing on refining its behavior based on offline data.
>
>
>
> > 【W2 & Q2】 Lacking of Strong teacher distillation
>
> This method is applicable when the tokenizer is consistent, as discussed in the Limitations section. We select Qwen3-30B-A3B-Thinking-2507 as the teacher model to distill into the Qwen3-0.6B student model. The distillation is performed exclusively on sequences with a length of **20K**.
>
> | Bench          | Qwen3-0.6B-Thinking (Original Model, 20K) | Qwen3-0.6B-Distilled (20K ) |
> | -------------- | ----------------------------------------- | --------------------------- |
> | AIME-24        | 11.25                                     | 12.19                       |
> | AIME-2025      | 18.44                                     | 20.73                       |
> | MATH-500       | 75.20                                     | 76.18                       |
> | AMC            | 49.38                                     | 51.33                       |
> | Olympaid-bench | 44.51                                     | 45.01                       |
>
> **Conclusion:** HPD applies to advanced models, as evidenced by the distilled model outperforming the officially released version.
>
> > 【W3 & Q3】In Section 5.4, HPD relies on offline trajectories for sampling, yet the on-policy data setting involves student-generated rollouts. It is unclear how HPD reconciles this mismatch in practice and whether the method can fully leverage on-policy feedback.
>
> The on-policy data setting in **Section 5.4** refers to using HPD/SFT on the offline trajectories first, and then applying OPD to further improve the model’s performance.
>
> > > How does HPD reconcile this mismatch in practice?
>
> When the SFT response is fed into the model, the pre-filling operation is quick, allowing us to derive all token logits from this response. Given these logits, we can use multinomial sampling to perform student sampling and record the sampled token index in the vocabulary as the student’s sampled label. For a more detailed explanation, refer to Algorithm 1 in our paper.
>
> > > Can HPD fully leverage on-policy feedback?
>
> In **Section 5.4** and **Table 5** of our paper, we observe that HPD alone can match or even outperform **SFT + OPD**, indicating that HPD does indeed leverage on-policy feedback effectively. Furthermore, in **Figure 3**, we perform an ablation study, which shows that on-policy feedback is crucial, particularly in the later stages of training.
>
> **We hope we have addressed your concern, and hope you can provide a better judgment of our work.**

---

> > ### Author Rebuttal · Reviewer_JzR7 · 2026-04-03
> >
> > Thank you for the detailed rebuttal, which has addressed my concerns in W1 and W3. However, I remain unconvinced by W2. The reported gains from Qwen3-0.6B-Thinking to Qwen3-0.6B-Distilled appear marginal compared to the improvements demonstrated in Table 2 (Qwen2.5 7B→1.5B/3B and LLaMA3 8B→1B/3B). I have the following follow-up questions:
> >
> > 1. **Sequence length**. Why was 20K chosen as the distillation length? Challenging benchmarks like AIME are typically better evaluated at longer contexts (e.g., 64K+), and the truncation may be suppressing the student's potential.
> >
> > 2. **Teacher-student gap**. The teacher Qwen3-30B-A3B-Thinking-2507 achieves ~85 on AIME25, yet the student only improves from 18.44 to 20.73. Could you explain why the performance gap remains so large despite distillation?
> >
> > 3. **Student model scale**. If the bottleneck is student capacity, why not evaluate on Qwen3-1.7B or Qwen3-4B, consistent with the scales used in the main experiments? As you claim, the hybrid distillation approach should not introduce prohibitive computational overhead.
> >
> > I would appreciate clarification on these points before updating my score.
> >
> >
> > ==================
> >
> > Update: My primary concerns have been addressed by the author rebuttal, and I have increased my score accordingly.

---

> > > ### Author Response · Authors · 2026-04-04
> > >
> > > ## 1. Sequence Length
> > >
> > > In the paper, we used **8k** as the target length for distillation. In the rebuttal, our goal is to further enhance distillation performance and validate the effectiveness of HPD, which is why we selected **20k** as the upper limit **based on our available computational resources**. For training costs, please refer to our response to Reviewer oGWS Q1.
> > >
> > > **While many works typically use 256, 512, and 1024 tokens [1][2][3].**
> > >
> > > **References:**
> > >
> > > [1] On-Policy Distillation of Language Models: Learning from Self-Generated Mistakes, https://arxiv.org/pdf/2306.13649
> > >
> > > [2] ABKD: Pursuing a Proper Allocation of the Probability Mass in Knowledge Distillation via α-β-Divergence, https://arxiv.org/abs/2505.04560
> > >
> > > [3] Rethinking Kullback-Leibler Divergence in Knowledge Distillation for Large Language Models, https://arxiv.org/pdf/2404.02657
> > >
> > > **Scaling to longer contexts for distillation is also viable and would yield possible benefits.** For evaluation length, please refer to the details below.
> > >
> > >
> > > ## 2. Teacher-Student gap
> > >
> > > We list the following possible reasons:
> > >
> > > (1) **Max Position Embedding:**
> > >
> > > Qwen3-30B-A3B-Thinking-2507 has a max position of **256k** (see [config.json](https://huggingface.co/Qwen/Qwen3-30B-A3B-Thinking-2507/blob/main/config.json)) and achieves AIME25 scores up to 85 using **81k tokens**, as you mentioned.
> > >
> > > In contrast, Qwen3-0.6B has a max position of **40k** (see [config.json](https://huggingface.co/Qwen/Qwen3-0.6B/blob/main/config.json)), which limits its ability to process this long evaluation length.
> > >
> > > (2) **Evaluation Length:**
> > >
> > > We further use **38K** tokens, which is **consistent with the Qwen3 technical report** ([see page 13, last sentence](https://arxiv.org/pdf/2505.09388)). We have selected the official report results from [Table 19](https://arxiv.org/pdf/2505.09388), as shown below:
> > >
> > > |         | Qwen3-0.6B-Original |
> > > | ------- | ------------------- |
> > > | AIME-24 | 10.70               |
> > > | AIME-25 | 15.10               |
> > >
> > > Our evaluation of the original model:
> > >
> > > | Avg@32  | Qwen3-0.6B-Original | Min_generation | Max_generation | Avg_generation |
> > > | ------- | ------------------- | -------------- | -------------- | -------------- |
> > > | AIME-24 | 11.35               | 2012           | 38912          | 16719          |
> > > | AIME-25 | 18.85               | 1546           | 38912          | 14919          |
> > >
> > > Our evaluation results for the distilled model are as follows
> > >
> > > | Avg@32  | Qwen3-0.6B-Distillation | Min_generation | Max_generation | Avg_generation |
> > > | ------- | ----------------------- | -------------- | -------------- | -------------- |
> > > | AIME-24 | **15.78**                   | 3385           | 38912          | 29112          |
> > > | AIME-25 | **24.17**                   | 3504           | 38912          | 26902          |
> > >
> > >
> > >
> > > **Conclusion:** In the **previous** evaluation, we used **20K** tokens, and the AIME25 score improved from **18.44** to **20.73**, a relative improvement of **12.42**%. When extended to **38K** tokens, the AIME25 score improved from **18.85** to **24.17**, a relative improvement of **28.22**%. **Moreover, the distilled version demonstrates a deeper thinking pattern.**
> > >
> > > Finally, we would like to note that evaluation on longer contexts is a systemic issue, influenced by factors such as the base model itself and its ability to handle long-form text.
> > >
> > >
> > > (3) Another possible factor is the student model parameters you mentioned, as outlined below.
> > >
> > >
> > >
> > > ## 3. Student model scale
> > >
> > > (1) We believe the student model is strong, and the performance could improve further. **Qwen3-4B-Thinking-2507,** which is an updated version of **Qwen3-4B-Thinking**, shows an improvement in AIME25 from **65.6** to **81.3**, a relative improvement of **23.93**%. In comparison, the relative improvement in our distillation for Qwen3-0.6B is **28.22**%, which demonstrates the effectiveness of HPD. https://huggingface.co/Qwen/Qwen3-4B-Thinking-2507
> > >
> > > (2) We conducted this experiment on Qwen3-0.6B to quickly validate your concerns regarding the more advanced teacher model mentioned in W2, and the results, including the extended thinking length, further consolidate its applicability. We believe HPD would work for Qwen3-1.7B or Qwen3-4B. In our paper, we also evaluate HPD on different student models at 1.5B and 4B.
> > >
> > > (3) Our distillation process indeed does not introduce prohibitive computational overhead compared to other distillation methods, **though training still requires computational resources. This is because we load 30B teacher models and still need to perform teacher-forward passes.** Please refer to our response to Reviewer oGWS
> > >  Q2 for further details. For Qwen3-0.6B, we trained for approximately 5 days on a single machine with 8 GPUs, completing about 5,000 steps (<1 epoch).
> > >
> > > ----
> > >
> > > **Thank you once again for your constructive comments.** We hope that further discussion will address this concern and that you will be able to raise your score.

---

### Official Review · Reviewer_E4qS · 2026-03-10

**Soundness:** 3
**Presentation:** 3
**Significance:** 3
**Originality:** 3
**Overall Recommendation:** 3
**Confidence:** 4

**Summary:**

This paper studies knowledge distillation for large language models through a unified perspective spanning three axes: divergence direction (forward vs. reverse KL), optimization strategy, and data regime (off-policy vs. on-policy). The authors proposes a reweighted log-likelihood view that casts several existing distillation objectives into a common framework. Building on this view, the authors introduce Hybrid Policy Distillation (HPD), a token-level method that combines forward- and reverse-KL-inspired signals with offline expert tokens and lightweight student-sampled tokens. Empirically, the method is evaluated on math reasoning, dialogue, and code tasks, under both off-policy and on-policy settings, and is reported to improve optimization stability, efficiency, and final downstream performance relative to several baselines.

In summary, think the paper has a good core idea and tackles an important problem, but it currently feels somewhat over-broad. It tries to contribute both a unifying conceptual framework and a new practical algorithm, and each of those would benefit from sharper presentation.

**Compliance With Llm Reviewing Policy:**

Affirmed.

**Final Justification:**

As discussing with the authors, I believe there is several missing baselines that try to combine FKL and RKL (in different ways) [1-4]. I understand these works are doing it in a different way. But the authors need to compare with them as they are highly related and also combining FKL and RKL. As these baselines are missing, i believe my original score did not reflect that and I will update it accordingly if.

[1] Ko, Jongwoo, et al. "Distillm-2: A contrastive approach boosts the distillation of llms." arXiv preprint arXiv:2503.07067 (2025). \
[2] Jung, Seongryong, et al. "Todi: Token-wise distillation via fine-grained divergence control." Proceedings of the 2025 Conference on Empirical Methods in Natural Language Processing. 2025. \
[3] Kim, Yeongmin, et al. "Distillation of Large Language Models via Concrete Score Matching." arXiv preprint arXiv:2509.25837 (2025). \
[4] Huang, Haiduo, et al. "SelecTKD: Selective Token-Weighted Knowledge Distillation for LLMs." arXiv preprint arXiv:2510.24021 (2025).

**Key Questions For Authors:**

- can the authors provide a one-sentence statement of the exact key novelty of HPD relative to prior hybrid KL / on-policy KD methods?

- How much of the gain comes from the sampled student token versus the hybrid divergence logic itself? For example, what happens if one uses only the expert-token branch, only the sampled-token branch, or a softer suppression mechanism?

**Limitations:**

yes

**Strengths And Weaknesses:**

**strengths:**
- Well-motivated problem formulation and the paper is nicely written.
- The reported empirical gains appear consistent across several settings.

**Weaknesses:**

- While the paper presents HPD as a new hybridization of forward/reverse KL and offline/online data regimes, in my opinion much of the method appears to combine existing ingredients: token-level reweighting, reverse-KL-based sampled-token correction, forward-KL-style expert reinforcement, and lightweight on-policy sampling. Hence, I think the paper would benefit from a sharper explanation of what is fundamentally new here beyond a particular engineering combination of known ideas

---

> ### Author Rebuttal · Authors · 2026-03-31
>
> > 【W1】 I think the paper would benefit from a sharper explanation of what is fundamentally new here, beyond a particular engineering combination of known ideas
>
> (1) We frame **distillation** as a **token-level reweighting scheme** to improve the understanding and design of new algorithms.
>
> (2) We combine the correction of RKLD and the reinforcement of FKLD to create our hybrid method. Additionally, the lightweight on-policy sampling introduced by our approach has not been seen before.
>
> (3) However, we would like to emphasize that HPD unifies them in a novel way to address key challenges in LLM distillation. First, we achieve the **efficiency of one-hot targets** similar to SFT while outperforming other baselines that rely on full or top-k distributions. Another key innovation is our **mask mechanism**, which utilizes a reverse-KL estimator to determine when to apply FKLD or RKLD, in contrast to methods that directly fuse them. Additionally, we employ **lightweight on-policy sampling** to effectively balance exploration and exploitation, eliminating the need for full-sequence rollouts.
>
> Therefore, we are not merely combining these methods; all these components are essential.
>
>
>
> > 【Q1】Can the authors provide a one-sentence statement of the exact key novelty of HPD relative to prior hybrid KL / on-policy KD methods?
>
>
>
> HPD combines one-hot target efficiency like SFT, token-level KL divergence integration with a reverse k1 estimator to determine when to apply FKL or RKL, and lightweight on-policy sampling to balance exploration and exploitation without full-sequence rollouts.
>
>
>
> > 【Q2】How much of the gain comes from the sampled student token versus the hybrid divergence logic itself? For example, what happens if one uses only the expert-token branch, only the sampled-token branch, or a softer suppression mechanism?
>
> > >  How much of the gain comes from the sampled student token versus the hybrid divergence logic itself? Only the expert-token branch?
>
> As shown in Section 6, Figure 3, allowing the student to sample the next token results in a performance increase compared to using only the hybrid divergence logic on the expert token. For the Qwen2.5-1.5B model, this approach leads to an improvement of nearly 3 points on the AIME benchmark, particularly in the later stages. This suggests that enabling student exploration is crucial for performance gains.
>
> > > only the sampled-token branch?
>
> Training using only the sampled-token branch, without the expert token, leads to instability and prevents the effective utilization of the teacher's knowledge, leading to negative optimization dominating the training process.
>
> > > softer suppression mechanism?
>
> The softer suppression mechanism is practical, but not gradient-accurate. We substitute $\log q - \log p$ with $q - p$, where $q$ is the teacher's predicted distribution and $p$ is the student's. To clearly demonstrate its effect, we show the training dynamics in OPD. The performance and KL loss are provided in the respective links (https://anonymous.4open.science/r/HPD-E273/softer_kl_koss.png and https://anonymous.4open.science/r/HPD-E273/softer_performance.png). We observe that $\log q - \log p$ yields a lower KL gap and better performance, indicating that gradient-accuracy results in real optimization.
>
> **We hope we have addressed your concern, and hope you can provide a better judgment of our work.**

---

> > ### Author Rebuttal · Reviewer_E4qS · 2026-04-03
> >
> > I thank the authors for their response.
> >
> > > HPD combines one-hot target efficiency like SFT, token-level KL divergence integration with a reverse k1 estimator to determine when to apply FKL or RKL, and lightweight on-policy sampling to balance exploration and exploitation without full-sequence rollouts.
> >
> > As the authors stated, one of the main contributions of this work is the integration of FKL and RKL without full-sequence rollouts. Now that I understand the paper more, I believe there is several missing baselines that try to combine FKL and RKL (in different ways) [1-4]. I understand these works are doing it in a different way. But the authors need to compare with them as they are highly related and also combining FKL and RKL. As these baselines are missing, i believe my original score did not reflect that and I will update it accordingly.
> >
> > [1] Ko, Jongwoo, et al. "Distillm-2: A contrastive approach boosts the distillation of llms." arXiv preprint arXiv:2503.07067 (2025). \
> > [2] Jung, Seongryong, et al. "Todi: Token-wise distillation via fine-grained divergence control." Proceedings of the 2025 Conference on Empirical Methods in Natural Language Processing. 2025. \
> > [3] Kim, Yeongmin, et al. "Distillation of Large Language Models via Concrete Score Matching." arXiv preprint arXiv:2509.25837 (2025). \
> > [4] Huang, Haiduo, et al. "SelecTKD: Selective Token-Weighted Knowledge Distillation for LLMs." arXiv preprint arXiv:2510.24021 (2025).

---

> > > ### Author Response · Authors · 2026-04-03
> > >
> > > Thank you for your detailed and thoughtful review. Before we dive into addressing your concerns, we would like to clarify some potential misunderstandings and provide a clearer explanation of our contributions.
> > >
> > > # Contribution
> > >
> > > (1) **HPD uses a policy gradient loss** **rather than relying on divergence as a loss function**. **It applies KL estimates as a weighted likelihood loss**, with gradient correctness as the guiding principle. The first three references you listed focus on designing divergence-based losses.
> > >
> > > (2) **One sentence is easy to confuse**; we have given the detailed description as follows:
> > >
> > > 1. **HPD incorporates one-hot target efficiency**, similar to SFT, enabling distillation on long trajectories, such as 10,240 tokens. Other methods that rely on full or top-k distributions struggle to achieve this.
> > >
> > > 2. **HPD employs a masking mechanism** to emphasize the student's underestimated expert tokens and suppress the overestimated ones, avoiding gradient contradictions found in methods that directly combine both terms, such as in your listed reference 2.
> > >
> > > 3. We use **lightweight on-policy sampling** during the **pre-filling phase** (where we sample the next token given a prefix) to balance exploration and exploitation, without the need for full-sequence rollouts during **offline distillation**.
> > >
> > > Therefore, **efficiently and effectively combining FKL and RKL is just one of the key advantages of this method.**
> > >
> > > # Baselines
> > >
> > > ## 1. Existing comparisons
> > >
> > > (1) In our paper, we use JSD as one of the baselines combining FKL and RKL, and we demonstrate that HPD performs better in the offline setting. **In our ablation study, we show that merely combining FKL and RKL is not effective during the later stages of training.**
> > >
> > > (2) During the rebuttal, we further compare to other combined FKL and RKL works as follows:
> > >
> > > **Baseline Efficiency**: We compare computational resources as follows: using Qwen2.5-7B as the teacher and Qwen2.5-1.5B as the student, with a batch size of 32, a sequence length of **10240**, and a vocabulary size of 152064. The memory for a single tensor at bfloat16 precision is 92.81 GiB.
> > >
> > > | Metric  | SFT  | HPD| AKL | ABKD  |
> > > | ---- | -- |  --- | --- | ------ |
> > > | Raw model-state memory  | ~22.35 GiB | ~35.39GiB   | ~35.39GiB  | ~35.39 GiB |
> > > | Extra logits| ~92.8GiB   | ~370.8 GB | ~2.2TiB | ~2.5TiB    |
> > > | Wall-clock (compared to SFT) | 1.0x  | ~2.3x -2.7x | ~3.5x - 5x | ~2.8x-3.3x |
> > >
> > > **Baseline Performance:**  For coding tasks, we use 4096 tokens for distillation.
> > >
> > > | Metric | SFT | ABKD | AKL | HPD  |
> > > | --- | -- | --- | --- | ---- |
> > > | HumanEval  | 73.80   | 74.40    | 75.00    | **79.30** |
> > > | MBPP | 67.70 | 73.50  | 75.40    | **75.40** |
> > > |Time | 3h20min | 14 hours | 27 hours | 7h30m |
> > >
> > > **HPD offers significant efficiency gains while delivering superior performance compared to other methods.**
> > >
> > > ## 2. Regarding the baselines you mentioned:
> > >
> > > 1. DistilLLM2 is based on an **on-policy setting** and can be considered an optimized method of OPD. In contrast, **we focus on offline distillation**, while HPD optimizes the SFT phase.
> > > 2. ToDi directly combines FKL and RKL using a sigmoid function: $D = a \cdot FKL + (1 - a) \cdot RKL),$ where $a = \sigma(\log p - \log q)$.
> > > 3. CSD enables more precise knowledge transfer by comparing the weighted and normalized logits of the teacher and student models, rather than directly comparing their softmax outputs. Because they think softmax smooths the logits into probabilities, potentially losing critical information about the relative magnitude of the logits. These logits are then **weighted** and **normalized**, and the **difference** between the teacher's and student's logits is used to calculate a **gradient**. **It does not involve combining FKL and RKL**.
> > > 4. SelecTKD is a **component mechanism** that can be added to existing methods to decide which token should undergo distillation. **It does not involve combining FKL and RKL**.
> > >
> > > **We acknowledge that only Reference 2 represents the combination of FKL and RKL in the offline setting.** It uses this combination as the **loss function for optimization**. The efficiency comparison is as follows:
> > >
> > > ### [Updated Results]
> > > | Metric  | HPD    | ToDi |
> > > | --- | ---- | ---- |
> > > | model-state memory | ~35.39GiB   | ~35.39GiB |
> > > | Extra logits  | **~370.8 GB**  | ~1.5 ~ 1.9TiB |
> > > | Wall-clock | ~2.3x -2.7x | ~2.7x ~ 3.1x  |
> > > | HumanEval |  **79.30** |  78.00  |
> > > | MBPP | **75.40** |   74.10 |
> > >
> > >
> > > ---
> > >
> > > **The experiment is expected to take approximately 22 hours, while our HPD only requires 7h30m, highlighting its efficiency.** The design of ToDi and AKL is similar, as both methods select the appropriate weighted terms to combine FKL and RKL. You can refer to the experiments above for more details.
> > >
> > > ---
> > >
> > > **Thank you once again for your constructive comments.** We hope that further discussion will address this concern and that you will be able to raise your score.

---

### Official Review · Reviewer_oGWS · 2026-03-12

**Soundness:** 4
**Presentation:** 3
**Significance:** 2
**Originality:** 3
**Overall Recommendation:** 4
**Confidence:** 4

**Summary:**

The paper first proposes a unified view of distillation as reweighted token-level log-likelihood optimization, where SFT, FKLD, and RKLD differ mainly in the data source and token weight function. It then introduces Hybrid Policy Distillation (HPD): for each offline expert token, it computes a reverse-KL-style gap and adds forward-KL reinforcement only when the student underestimates the expert token; separately, it samples a non-expert token from the student under the same offline prefix and suppresses it when that sampled token appears overestimated relative to the teacher. The final loss applies both weights jointly, so HPD mixes forward-KL support for expert tokens with reverse-KL-style suppression of bad student tokens, while retaining one-hot-like efficiency.

Experimentally, the paper evaluates off-policy reasoning distillation on Qwen2.5 and LLaMA3 students using OpenR1-Math-8192, plus dialogue distillation on UltraFeedback prompts and code distillation on WizardCoder-style data. The reasoning results are the strongest part: for example, in the off-policy setting HPD improves the Qwen 7B to 3B average from 32.67 (SFT) to 39.83 and the LLaMA 8B to 3B average from 29.45 to 34.56; in the on-policy continuation setting, HPD initialization plus OPD also outperforms SFT plus OPD. Dialogue and code gains are directionally positive but smaller.

**Compliance With Llm Reviewing Policy:**

Affirmed.

**Final Justification:**

well motivated paper despite solution used many well understood common techniques. results are encouraging.

**Key Questions For Authors:**

Can you add direct comparisons against DistiLLM, ABKD/AKL, and at least one stronger on-policy/hybrid baseline such as MiniLLM, GKD, or DistiLLM-2? Right now the empirical positioning against the state of the art is the biggest gap.

What is the exact end-to-end compute cost of HPD relative to SFT, SeqKD, and SFT→OPD? Please report training-time wall-clock, teacher forward passes, and memory.

Did you perform decontamination / overlap checks between OpenR1-Math-8192 and the reasoning benchmarks? Please state this explicitly.

Why is the reinforce rule in Equation (14) specifically a doubling of the forward-KL weight? Is this derived, tuned, or arbitrary? A sensitivity study would help.

In the on-policy section, why are MiniLLM/GKD not shown as full direct baselines in the final comparison table, despite being discussed in setup?

Can you report multi-seed variance/confidence intervals more broadly, especially for personalization and coding, not just significance markers on the reasoning table?

**Limitations:**

yes

**Strengths And Weaknesses:**

Strength:
------
The paper is well motivated. The unified “reweighted likelihood” perspective is a clean way to organize several distillation objectives, and the token-level explanation for why negative weights suppress one token while redistributing mass to alternatives is intuitive and helpful. Even if not theoretically deep, it makes the design space easier to reason about.

The empirical coverage is broader than average for a distillation submission. The authors test across two model families, multiple student scales, long-form reasoning, short-form dialogue, code generation, and an OPD follow-up stage. The reasoning improvements are not trivial; they are material enough that I believe the method is doing something real, especially for the 3B students.

I also like that the method is computationally plausible. Compared with full-distribution FKLD over large vocabularies, HPD uses token-level signals on expert and sampled tokens only. The ablation story is also sensible: the paper argues that student sampling helps avoid premature convergence and that the reinforce operation stabilizes alignment.


Weakness:
------
The main issue is novelty. The paper’s framing suggests a fairly new joint treatment of divergence direction, optimization strategy, and data regime, but in practice much of this territory is already occupied. MiniLLM already makes the reverse-KL case for LLM distillation; GKD already leverages student on-policy trajectories; DistiLLM already mixes a nonstandard KL objective with adaptive use of student-generated outputs; DistiLLM-2 already uses both teacher and student responses in one contrastive distillation framework; and ABKD / AKL already study adaptive interpolation between forward- and reverse-KL effects. HPD’s contribution is therefore the specific token-level masking and reinforcement rule, not the broader hybridization story. That feels incremental rather than a clear conceptual jump.

A related issue is baseline strength. For off-policy reasoning, the paper compares against SFT, SeqKD, RKLD, and JSD, but omits stronger modern baselines such as DistiLLM, ABKD, and AKL. For on-policy reasoning, the setup discusses MiniLLM and GKD, but the headline table mainly compares SFT, SFT→OPD, HPD, and HPD→OPD, which makes the evaluation feel more like “better initialization for OPD” than a decisive comparison against the best available on-policy or hybrid distillation methods. Given how central the paper’s “hybrid” claim is, this omission matters a lot.

---

> ### Author Rebuttal · Authors · 2026-03-31
>
> > 【W1】The main issue is the novelty, given the presence of existing works.
>
> (1) We frame **distillation** as a **token-level reweighting scheme** to improve the understanding and design of new algorithms, while other works focus on divergence design.
>
> (2) Our work introduces several novel contributions compared to existing methods. First, we achieve the **efficiency of one-hot targets** similar to SFT while outperforming other baselines that rely on full or top-k distributions. Another key innovation is our **mask mechanism**, which utilizes a reverse-KL estimator to determine when to apply FKLD or RKLD, in contrast to methods that directly fuse them. Additionally, we employ **lightweight on-policy sampling** to effectively balance exploration and exploitation, eliminating the need for full-sequence rollouts.
>
> > 【W2】A related issue is baseline strength.
>
> See Q1 and Q3
>
> > [Q1] Can you add direct comparisons against DistiLLM, ABKD/AKL, and at least one stronger on-policy/hybrid baseline such as MiniLLM, GKD, or DistiLLM-2?
>
> Related papers use response lengths from 256 to 1024 tokens. **Our work is the first to use 8192 tokens, achieving both efficiency and high performance.**
>
> **(1) Baseline Efficiency**: We compare computational resources as follows: using Qwen2.5-7B as the teacher and 1.5B as the student, with a batch size of 32, a sequence length of 10240, and a vocabulary size of 152064. The memory for a single tensor at bfloat16 precision is 92.81 GiB.
>
> | Metric | SFT | HPD | AKL | ABKD |
> | ---------------------------- | ---------- | ----------- | ---------- | ---------- |
> | Raw model-state memory  | ~22.35 GiB | ~35.39GiB   | ~35.39GiB  | ~35.39 GiB |
> | Extra logits  | ~92.8GiB   | ~370.8 GB   | ~2.2TiB    | ~2.5TiB    |
> | Wall-clock | 1.0x | ~2.3x -2.7x | ~3.5x - 5x | ~2.8x-3.3x |
>
> **(2) Baseline Performance:**  For coding tasks, we use 4096 tokens for distillation.
>
> | Metric | SFT | ABKD | AKL| HPD |
> | ---------- | ------- | -------- | -------- | --------- |
> | HumanEval | 73.80| 74.40  | 75.00    | **79.30** |
> | MBPP| 67.70| 73.50  | 75.40| **75.40** |
> | Train Time | 3h20min | 14 hours | 27 hours | 7h30m|
>
> **HPD offers significant efficiency gains while delivering superior performance compared to other methods.**
>
> > 【Q2】What is the exact end-to-end compute cost of HPD relative to SFT, SeqKD, and SFT→OPD?
>
> || SFT | SeqKD | HPD  | OPD  |
> | ----------- | ---------- | ----------- | ----------- | ---------- |
> | training-time wall-clock | 1.0x   | 2.3x ~ 2.7x | 2.3x ~ 2.7x | >> 2.6x ~ 3.0x |
> | teacher forward passes   | 0 | 1| 1| 1|
> | raw model-state memory  | ~22.35 GiB | ~35.39 GiB  | ~35.39 GiB  | ~35.39GiB |
> | Rollout Time | Prefilling | Prefilling | Prefilling  | Decoding |
>
> **HPD avoids the time-consuming rollouts of OPD while offering better training efficiency.**
>
> > 【Q3】decontamination report
>
> We applied light and aggressive normalization for exact matches, followed by 6-gram candidates and SequenceMatcher for near-duplicate detection, with Math500 showing 16 high-confidence and 16 possible variant rows, and OlympiadBench showing 96 high-confidence rows. All other datasets showed zero results.
>
> > 【Q4】Why is the reinforcement rule in Equation (14) specifically a doubling of the forward-KL weight?
>
> The reinforcement rule is theoretically derived and tuned to align the teacher distribution with the ground truth tokens, based on the gradient of the FKLD loss. A heavy weight could lead to overfitting, as indicated by lower entropy, while our goal is to ensure the student closely aligns with the teacher.
>
> > 【Q5】In the on-policy section, why are MiniLLM/GKD not shown as full direct baselines in the final comparison table, despite being discussed in the setup?
>
> You can think of OPD as GKD with reverse KL (see Section 4, the last sentence of GKD variants paragraph on page 5 of (https://arxiv.org/pdf/2306.13649)). **GKD is a paradigm, while OPD is a specific implementation**. MiniLLM can be considered as SFT + OPD without teacher-mixed sampling (to avoid reward hacking and length normalization, as described in (https://arxiv.org/pdf/2306.08543), Section 2.3). Additionally, since we **focus on offline distillation**, we compare HPD, HPD + OPD, and SFT + OPD for fairness in the final performance comparison.
>
> > 【Q6】 Can you report multi-seed variance/confidence intervals more broadly, especially for personalization and coding?
>
> Coding tasks are evaluated based on greedy decoding. We report CI intervals of the alignment task as follows:
>
> | Dataset | Arena-Hard | 95% CI |
> | ------- | ---------- | ----------- |
> | Base | 7.47  | (-1.0, 1.1) |
> | SFT | 18.10 | (-1.5, 1.5) |
> | SeqKD | 15.40   | (-1.2, 1.2) |
> | RKLD | 17.80  | (-1.1, 1.0) |
> | JSD  | 20.20 | (-1.5, 1.6) |
> | HPD  | 21.80 | (-0.9, 1.2) |
>
> HPD achieves a low CI interface, demonstrating its stability.
>
> ---
>
> **Thank you for your excellent and expert review! We hope we have addressed your concern, and hope you can provide a better judgment of our work.**

---

> > ### Author Rebuttal · Reviewer_oGWS · 2026-04-03
> >
> > author responded to my previous question and explained the novelty.

---

> > > ### Author Response · Authors · 2026-04-05
> > >
> > > Thank you for recognizing our efforts and for your supportive comments. We are glad that our explanations have addressed your concerns satisfactorily.

---

### Decision · Program_Chairs · 2026-04-30

**Decision:**

Accept (regular)

**Comment:**

This paper studies knowledge distillation for LLMs and proposes HPD, a hybrid token-level distillation framework that combines forward- and reverse-KL signals with lightweight on-policy sampling. The paper was initially borderline, but the rebuttal addressed several important concerns, and Reviewer FeSF also indicated a willingness to raise the score, although this was not reflected in the displayed ratings. Overall, the reviewers found the problem important, the method practically appealing, and the empirical results encouraging across multiple tasks. While some concerns remained about novelty and baseline coverage, I find the overall contribution sufficient for acceptance.